



# Low Cobalt Inventories in the Amundsen and Ross Seas Driven by High Demand for Labile Cobalt Uptake Among Native Phytoplankton Communities

**Rebecca J. Chmiel[1,2], Riss M. Kellogg[1,2], Deepa Rao[1,2], Dawn M. Moran[2], Giacomo R. DiTullio[3], and Mak A. Saito[2]**

[1] MIT/WHOI Joint Program in Oceanography/Applied Ocean Science and Engineering, Woods Hole, MA, 02543, USA

[2] Department of Marine Chemistry and Geochemistry, Woods Hole Oceanographic Institution, Woods Hole, MA, 02543, USA

[3] Hollings Marine Laboratory, 331 Fort Johnson, Charleston SC, 29412, USA

Corresponding author: Mak Saito (msaito@whoi.edu)

**Key Points:**

- A significantly smaller dCo inventory was observed in the Ross Sea during the 2017/2018 austral summer compared to two expeditions in 2005/2006.

- The drawdown of the labile dCo fraction can be explained by higher rates of Co uptake by phytoplankton.

- This change may be due to the alleviation of Fe limitation through inputs from increased glacial melting and subsequent development of intermittent vitamin $B_{12}$ and/or Zn limitation, both of which would be expected to increase the demand for Co among plankton communities.



**Abstract**

Cobalt (Co) is a scarce but essential micronutrient for marine plankton in the Southern Ocean and coastal Antarctic seas where dissolved cobalt (dCo) concentrations can be extremely low. This study presents total dCo and labile dCo distributions measured via shipboard voltammetry in the Amundsen Sea, Ross Sea and Terra Nova Bay during the CICLOPS (Cobalamin and Iron Co-Limitation of Phytoplankton Species) expedition. A significantly smaller dCo inventory was observed during the 2017/2018 CICLOPS expedition compared to two 2005/2006 expeditions to the Ross Sea conducted over a decade earlier. The dCo inventory loss (~10–20 pM) was present in both the surface and deep ocean and was attributed to the loss of labile dCo, resulting in the near-complete complexation of dCo by strong ligands in the photic zone. A changing dCo inventory in Antarctic coastal seas could be driven by the alleviation of iron (Fe) limitation in coastal areas where the flux of Fe-rich sediments from melting ice shelves and deep sediment resuspension may have shifted the region towards vitamin $B_{12}$ and/or zinc (Zn) limitation, both of which are likely to increase the demand for Co among marine plankton. High demand for Zn by phytoplankton can result in increased Co and cadmium (Cd) uptake because these metals often share the same metal uptake transporters. This study compared the magnitudes and ratios of Zn, Cd and Co uptake ($\rho$) across upper ocean profiles and observed order of magnitude uptake trends ($\rho Zn > \rho Cd > \rho Co$) that paralleled the trace metal concentrations in seawater. High rates of Co and Zn uptake were observed throughout the region, and the speciation of available Co and Zn appeared to influence trends in dissolved metal : phosphate stoichiometry and uptake rates over depth. Multi-year loss of the dCo inventory throughout the water column may be explained by an increase in Co uptake into particulate organic matter (POM) and subsequently increased flux of Co into sediments via sinking and burial. This perturbation of the Southern Ocean Co biogeochemical cycle could signal changes in the nutrient limitation regimes, phytoplankton bloom composition, and carbon sequestration sink of the Southern Ocean.

**Plain Language Summary**

Cobalt is an important micronutrient for plankton, yet is often scarce throughout the oceans. A 2017/2018 expedition to coastal Antarctica, including regions of the Amundsen Sea and the Ross Sea, discovered lower concentrations of cobalt compared to two past expeditions in 2005 and 2006. In particular, this expedition observed lower concentrations of deep-ocean labile cobalt, or "free" cobalt unbound to strong organic molecules, the type of cobalt preferred by phytoplankton for uptake as a micronutrient. It is possible that a shifting nutrient landscape due to changing inputs of other micronutrients like dissolved iron is causing the lower dissolved cobalt concentrations, and may also be affecting the demand for micronutrients like dissolved zinc and vitamin $B_{12}$, which contains a cobalt atom. We have modeled how increased cobalt uptake by plankton can result in the lower deep cobalt concentrations over a time period of 12 years.

**1 Introduction**

Coastal Antarctic seas are highly productive environments for phytoplankton blooms and are characterized by high nutrient, low chlorophyll (HNLC) surface waters that tend to be growth limited by iron (Fe) and other trace metal micronutrients (Martin et al., 1990; Arrigo et al., 2008, 2012). During the spring and summer months, katabatic winds and fragmenting sea ice form open coastal polynyas in the Amundsen and Ross Seas that host high phytoplankton productivity and act as significant global carbon sinks (Arrigo et al., 2012). In the winter, ice cover supports the



turnover of deep waters that allow trace metals like Fe to be redistributed to the upper ocean (Sedwick and DiTullio, 1997; Sedwick et al., 2011). Phytoplankton blooms in coastal Antarctic polynyas are dominated by eukaryotes such as diatoms and the haptophyte *Phaeocystis antarctica* (Arrigo et al., 1999; DiTullio et al., 2003), while cyanobacteria like *Prochlorococcus* and *Synechococcus*, which are highly abundant in the adjacent South Pacific and South Atlantic gyres, are near-absent from the phytoplankton community in the Southern Ocean (DiTullio et al., 2003; Bertrand et al., 2011; Chandler et al., 2016).

Cobalt (Co) is an essential trace metal nutrient for many marine plankton and is relatively scarce in the marine environment, often present in the dissolved phase (dCo) in picomolar concentrations ($10^{-12}$ mol $L^{-1}$). Co acts as a cofactor for metalloenzymes like carbonic anhydrase, a crucial enzyme in the carbon concentrating mechanism of photosynthetic phytoplankton (Sunda and Huntsman, 1995; Roberts et al., 1997; Kellogg et al., 2020), and vitamin $B_{12}$ (cobalamin), which can be used for the biosynthesis of methionine but is only produced by some bacteria and archaea (Warren et al., 2002; Bertrand et al., 2013). In the Ross Sea, vitamin $B_{12}$ availability has been observed to co-limit phytoplankton growth with iron (Fe) when bacterial abundance is low (Bertrand et al., 2007). Some phytoplankton exhibit flexible vitamin $B_{12}$ metabolisms and can express a vitamin $B_{12}$-independent methionine synthase pathway (*metE* gene) instead of the vitamin $B_{12}$-dependent pathway (*metH* gene), allowing these organisms to thrive in vitamin-depleted environments (Rodionov et al., 2003; Bertrand et al., 2013; Helliwell, 2017). Recently, *P. antarctica* was discovered to contain both *metH* and a putative *metE* gene, displaying a metabolism that is flexible to vitamin $B_{12}$ availability (Rao et al., [In review]). Additionally, recent observations of Zn co-limitation with Fe have been documented in the Ross Sea (Kellogg et al., [Submitted]), suggesting a complex landscape of trace metal and vitamin stress interactions in the otherwise macronutrient-rich waters of coastal Antarctica.

Dissolved Co is present as two primary species in the marine environment: a "free" labile Co(II) species with weakly bound ligands and a Co(III) species that is strongly bound to organic ligands ($K_s > 10^{16.8}$) (Saito et al., 2005). Labile dCo is considered to be more bioavailable to marine microbes than strongly-bound dCo, although there is evidence that phytoplankton communities can access Co in strongly-bound organic ligand complexes (Saito and Moffett 2001) and that microbial communities may produce extracellular Co ligands that stabilize dCo and prevent its loss via scavenging to manganese (Mn)-oxide particles (Saito et al., 2005; Bown et al., 2012). Previous dCo sampling expeditions to the Ross Sea, including two 2005/2006 Controls of Ross Sea Algal Community Structure (CORSACS) expeditions (Saito et al., 2010) and fieldwork in 2009 that sampled the water column below early spring sea ice in the McMurdo Sound (Noble et al., 2013), reported relatively high concentrations of labile dCo in the surface Ross Sea when compared to the tropical and subtropical global oceans, suggesting that labile dCo was fairly replete and bioavailable to phytoplankton at the time (Saito et al., 2010).

This study examines the biogeochemical cycle of Co in the Amundsen and Ross Seas during the 2017/2018 austral summer as part of the Cobalamin and Iron Co-Limitation of Phytoplankton Species (CICLOPS) expedition. Here, we present profiles of dCo speciation that revealed a lower dCo inventory during the 2017/2018 summer bloom compared to that observed during the 2005/2006 CORSACS expeditions, as well as mostly undetectable concentrations of labile dCo in the surface ocean. Additional datasets of dissolved zinc (dZn) and cadmium (dCd), as well as profiles of Co, Zn and Cd uptake rates measured by isotope tracer incubation experiments suggest that regions of vitamin $B_{12}$ and Zn stress within phytoplankton blooms could



be driving high demand for bioavailable Co in the surface ocean. The results presented by this
study reveal a substantial perturbation of the Co cycle and a shift towards vitamin $B_{12}$ and/or Zn
limitation in coastal Antarctic waters impacted by high rates of glacial ice melt and a warming
climate.

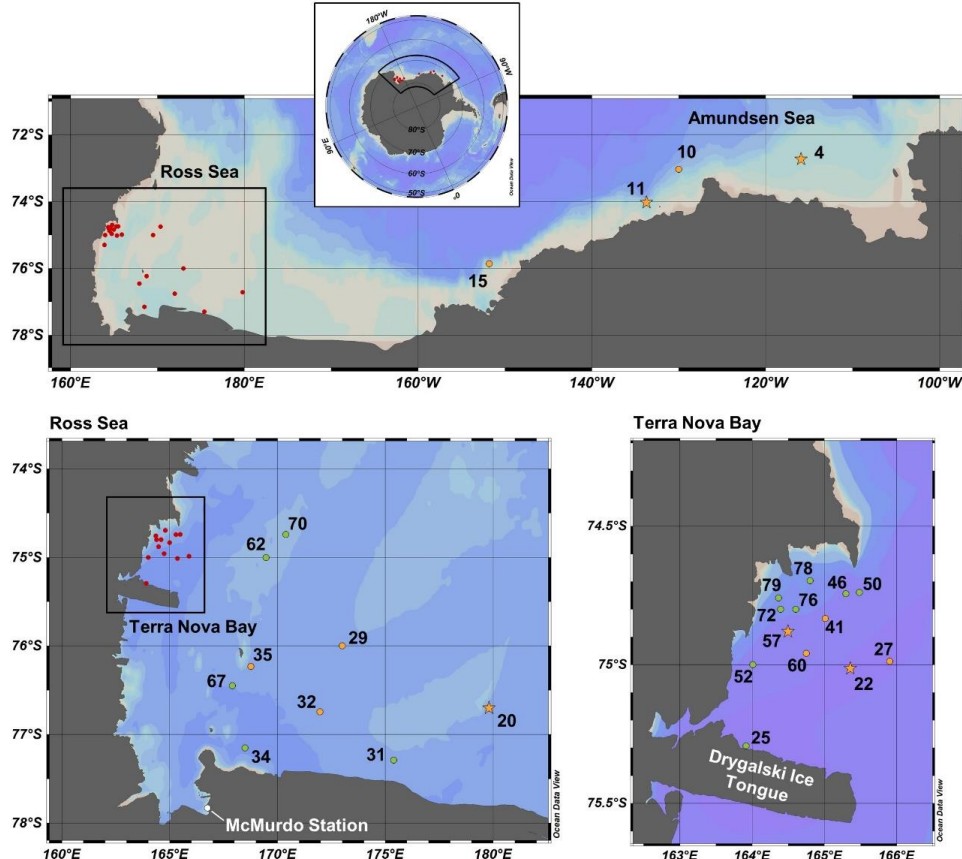

**Figure 1.** Map of CICLOPS stations in coastal Antarctic waters, including insets of stations within
the Ross Sea and Terra Nova Bay. Dissolved Co, dZn and dCd were analyzed at stations marked
in yellow, and stations marked in green were analyzed for dZn and dCd, but electrochemical dCo
measurements were not conducted. At stations marked with a star, Co, Zn and Cd uptake profiles
are presented in this study. Stations marked in red are shown in more detail in an inset. Note that
the grey coastline marks both terrestrial coastline and areas of consistent ice, including ice shelves
and glaciers; this includes the Drygalski Ice Tongue, a glacier to the south of Terra Nova Bay.
**2 Methods**
2.1 Study area and trace metal sampling
Samples were collected along the coastal Antarctic shelf from the Amundsen Sea, Ross
Sea, and Terra Nova Bay (Fig. 1) during the CICLOPS expedition on the RVIB *Nathanial B.*



*Palmer* (NBP-1801; December 16, 2017 – March 3, 2018). Dissolved seawater was collected from full-depth station profiles using a trace metal clean sampling rosette deployed on a conducting synthetic line supplied by the U.S. Antarctic Program (USAP) and equipped with twelve 8 L X-Niskin bottles (Ocean Test Equipment) supplied by the Saito laboratory. Real-time trace metal rosette operations allowed for the careful collection of seawater from 10 and 20 m above the ocean floor to study sediment-water interactions within a potential nepheloid layer. After deployment, the X-Niskin bottles were transported to a trace metal clean van and pressurized with high-purity (99.999 %) $N_2$ gas. Seawater samples for nutrients, dCo and trace metal analysis were then filtered through acid-washed 0.2 µM Supor polyethersulfone membrane filters (Pall Corporation, 142 mm diameter) within 3 hours of rosette recovery.

To minimize metal contamination of samples, all sample bottles were prepared using trace metal clean procedures prior to the expedition. The cleaning procedure for dCo sample bottles entailed soaking sample bottles for ~1 week in Citranox, an acidic detergent, rinsing with Milli-Q water (Millipore), soaking sample bottles for ~2 weeks in 10% trace metal grade HCl (Optima, Fisher Scientific), and rinsing with lightly acidic Milli-Q water (< 0.1% HCl). Macronutrient sample bottles were rinsed with Milli-Q water and soaked overnight in 10% HCl. The procedure for total dissolved metal sample bottles (dZn and dCd) was identical to that used for dCo bottles except the Citranox soak step was omitted.

Samples for dCo analysis were collected in 60 mL low-density polyethylene (LDPE) bottles and stored at 4ºC until analysis. Duplicate dCo samples were collected: one for at-sea analysis of labile dCo and total dCo, and another for preservation and total dCo analysis in the laboratory after the expedition. Preserved total dCo samples were stored with oxygen-absorbing satchels (Mitsubishi Gas Chemical, model RP-3K), which preserve the sample for long-term storage and future analysis (Noble et al., 2017; Bundy et al., 2020). Preserved dCo samples were stored in groups of 6 within an open (unsealed) plastic bag, which was then placed into a gas-impermeable plastic bag (Ampac) with one oxygen-absorbing satchel per 60 mL dCo sample. The outer bag was then heat-sealed and stored at 4ºC until analysis.

Samples for total dissolved metal analysis (dZn and dCd) were collected in 250 mL LDPE bottles and stored double-bagged at room temperature. After ~7 months, the total dissolved metals samples were acidified to a pH of 1.7 with trace metal grade HCl (Optima, Fisher Scientific), and were stored acidified for more than one year before instrumental analysis.

2.2 Dissolved Co and labile dCo analysis

Total dCo – the combined fractions of labile and ligand-bound dCo, hereafter simply dCo – and labile dCo concentrations were analyzed via cathodic stripping voltammetry (CSV) as described by Saito and Moffett (2001) and modified by Saito et al. (2010) and Hawco et al. (2016). CSV analysis was conducted using a Metrohm 663 VA and µAutolabIII systems equipped with a hanging mercury drop working electrode. All reagents were prepared as described in Chmiel et al. (2022). Most samples were analyzed at sea within 3 weeks of sample collection, and stations 57 and 60 were analyzed for labile dCo at sea and their duplicate preserved samples were analyzed for total dCo in November 2019 in the laboratory.

To measure total dCo concentrations, filtered seawater samples were first UV-irradiated in quartz tubes for one hour in a Metrohm 705 UV Digester to destroy natural ligand-bound Co complexes. 11 mL of sample was then added to a 15 mL trace metal clean polypropylene vial, and



100 µL of 0.1 M dimethyglyoxime (DMG; Sigma Aldrich) ligand and 130 µL of 0.5 M N-(2-
hydroxyethyl)piperazine-N-(3-propanesulfonic acid) (EPPS, Sigma Aldrich) buffer was added to
each sample vial. A Metrohm 858 Sample Processor then loaded 8.5 mL of each sample into the
electrode's Teflon cup and added 1.5 mL of 1.5 M $NaNO_2$ reagent (Merck). The mercury electrode
performed a fast linear sweep from -1.4 V to -0.6 V at a rate of 5 V s$^{-1}$ and produced a cobalt
reduction peak at -1.15 V, the voltage at which the $Co(DMG)_2$ complex is reduced from Co(II) to
Co(0) (Saito and Moffett, 2001). The height of the Co reduction peak is linearly proportional to
the amount of total dCo present in the sample. Peak heights were determined by NOVA 1.10
software. A standard curve was created with 4 additions of 25 pM dCo to each sample, and a type-
I linear regression of the standard addition curve performed by the LINEST function in Microsoft
Excel allowed for the calculation of the initial amount of Co present in the sample.

When analyzing labile dCo concentrations, samples were not UV-irradiated so as to only
quantify the free or weakly bound dCo not bound to strong organic ligands. 11 mL of labile
samples were instead allowed to equilibrate with the DMG ligand and EPPS reagent overnight (~8
hours) before analysis to allow time for the labile dCo present in the sample to bind to the DMG
ligand via competitive ligand exchange (K > $10^{16.8}$). Labile dCo samples were then loaded onto
the Sample Processor and analyzed electrochemically using identical methods as described above
for total dCo samples.
2.3 Dissolved Co standards and blanks

During the CICLOPS expedition, an internal standard consisting of filtered, UV-irradiated
seawater was analyzed for dCo every few days while samples were being analyzed (39 ± 4 pM, n
= 9). While additional preserved dCo samples were analyzed in the laboratory in November 2019,
triplicate GSC2 GEOTRACES community intercalibration standards were carefully neutralized to
a pH of ~8 using negligible volumes of ammonium hydroxide ($NH_4OH$) and analyzed for dCo.
This is the same intercalibration batch originally reported in Table 1 of Chmiel et al. (2022), as
analysis for both expeditions overlapped temporally. The GSC2 standard was determined to have
a dCo concentration of 80.2 ± 6.2 (n = 3), a value that is very similar to the one reported by Hawco
et al., (2016) (77.7 ± 2.4). Currently, no official community consensus for dCo in the GSC2
intercalibration standard exists.

Analytical blank measurements for each reagent batch (a unique combination of DMG,
EPPS, and $NaNO_2$ reagent batches) were measured to determine any Co contamination due to
reagent impurities. Blanks were prepared in triplicate with UV-irradiated surface seawater passed
through a column with Chelex 100 resin beads (Bio-Rad) to remove metal contaminants, then UV-
irradiated again. Chelex beads were prepared as described in Price et al. (2013) to remove organic
impurities from leaching into the eluent. For the 5 batches of reagents used on this expedition, the
analytical blanks were found to be 2.3 pM, 4.0 pM, 10.1 pM, 15.6 pM, and 8.6 pM dCo, with an
average of 8.1 pM Co. The analytical blank detected for the laboratory-run total dCo samples was
1.0 pM. It should be noted that blank values above 10 pM are considered high for this method.
Analytical blank values were subtracted from the measured Co values determined with the
respective reagent batch. The average standard deviation within each triplicate batch of blanks (1.3
pM) was used to estimate the analytical limit of detection (3 × blank standard deviation) of 4 pM.
When detectable dCo concentrations were found below the 4 pM detection limit, their values were
preserved in the dataset and flagged as below the detection limit (<DL). In cases where no dCo or
labile dCo were detected (i.e., when no peak was measurable and/or the dCo value predicted was



$< 0$ pM), values of 0 pM were assigned for the purposes of plotting and selecting statistical analysis
and were flagged as not detected (n.d.) as well as <DL in the dataset; although these concentrations
were not detectable with our methodology, we believe the incredibly low concentrations of dCo
and labile dCo observed on this expedition were meaningful, and that removing these values from
our analysis misrepresents the data and would skew the results to appear higher than was observed.
**Table 1.** Mean dCo and labile dCo values measured in the surface ocean (10 m) and the deep
ocean (> 100 m) in the three regions sampled. One dCo sample and numerous labile dCo samples
were determined to be below the analytical detection limit (<DL) of 4 pM. Only using the values
measured above the detection limit would artificially inflate the calculation of the mean value;
instead, samples measured between 0 and the DL were left unaltered as their originally measured
value and samples with no detected concentrations of dCo or labile dCo (n.d.) were adjusted to 0
pM. The number of samples included in the mean calculation that are <DL is indicated by $n_{<DL}$.

| | | Surface (10 m) | | | |
|---|---|---|---|---|---|
| Region | n | $dCo_{mean}$ [pM] | $n_{<DL}$ for dCo | Labile $dCo_{mean}$ [pM] | $n_{<DL}$ for labile dCo |
| Amundsen Sea | 4 | $28 \pm 7$ | 0 | $5 \pm 6$ | 2 |
| Ross Sea | 4 | $28 \pm 12$ | 0 | $1 \pm 2$[a] | 4 |
| Terra Nova Bay | 5 | $11 \pm 7$ | 1 | n.d.[b] | 5 |
| | | Deep (> 100 m) | | | |
| Region | n | $dCo_{mean}$ [pM] | dCo $n_{<DL}$ | Labile $dCo_{mean}$ [pM] | Labile dCo $n_{<DL}$ |
| Amundsen Sea | 30 | $41 \pm 5$ | 0 | $4 \pm 4$ | 14 |
| Ross Sea | 32 | $46 \pm 8$ | 0 | $9 \pm 7$ | 9 |
| Terra Nova Bay | 34 | $39 \pm 18$ | 0 | $6 \pm 8$ | 18 |

[a] Of the 4 surface samples analyzed for labile dCo in the Ross Sea, 3 were n.d. and the fourth
contained 3.5 pM labile dCo.
[b] All surface samples in Terra Nova Bay were n.d. for labile dCo.
2.4 Dissolved Zn and Cd analyzed by ICP-MS
Total dissolved trace metal samples were analyzed for dZn and dCd using isotope dilution
and inductively coupled plasma mass spectrometry (ICP-MS) as described in Kellogg et al.
(Submitted) based on methodology described in Cohen et al. (2021). Briefly, 15 mL of acidified
filtered seawater samples were spiked with an acidified mixture of stable isotopes including $^{67}$Zn,
and $^{110}$Cd, among other metal stable isotopes, and pre-concentrated via a solid phase extraction
system seaFAST-pico (Elemental Scientific) to an elution volume of 500 μL. The samples were
then analyzed using an iCAP-Q ICP-MS (Thermo Scientific) and concentrations were determined
using a multi-elemental standard curve (SPEX CertiPrep).
2.5 Co, Zn and Cd uptake rates via isotope incubations
Co, Zn and Cd uptake rates were quantified using incubations of collected marine microbial
communities spiked with stable or radioisotopes to trace the conversion of dissolved trace metal
into the particulate phase. Briefly, unfiltered seawater used for the incubation uptake experiments
was collected from the trace metal rosette, and the Co, Zn and Cd uptake incubations were spiked





with 0.1 pM $^{57}CoCl_2$, 2 nM $^{67}ZnO$ and 300 pM $^{110}CdO$, respectively. All incubation bottles were
then sealed and placed in a flow-through shipboard incubator for 24 hours. The incubator was
shielded by black mesh screening to allow 20% ambient light penetration. Incubation biomass was
collected by vacuum filtration onto acid-rinsed 3 μm Versapor filters (Pall). The $^{57}Co$ incubation
filters were stored at room temperature in Petri dishes prior to radiochemical gamma-ray counting
both at sea and in the laboratory, and the $^{67}Zn$ and $^{110}Cd$ incubation filters were stored at -80 ⁰C in
acid-rinsed cryovials until ICP-MS analysis in the laboratory. See Kellogg et al. (Submitted), Rao
2020 and Kellogg (2022) for full methodology and instrumental analysis.
2.6 Pigment and phosphate analysis
Phytoplankton pigment samples were collected via filtration and analyzed for select
pigments by high-performance liquid chromatography (HPLC) as described in DiTullio and
Geesey (2003). Macronutrient samples were collected from the trace metal rosette alongside dCo
samples and were filtered using the same methodology as dCo and total metal samples (see above).
Samples were collected in 60 mL high-density polyethylene (HDPE) bottles and were stored
frozen until analysis. Dissolved $PO_4$ concentrations were determined by Joe Jennings at Oregon
State University via the molybdenum blue method (Bernhardt and Wilhelms, 1967) using a
Technicon AutoAnalyzer II attached to an Alpkem autosampler.
2.7 Historical dCo and pigment data
In this study, dCo profiles from the CICLOPS expedition are compared to those from
previous fieldwork in the Ross Sea, including the Controls of Ross Sea Algal Community Structure
(CORSACS) expeditions: CORSACS-1 (NBP-0601; December 27, 2005 – January 23, 2006) and
CORSACS-2 (NBP-0608; November 8, 2006 – December 3, 2006), reported in Saito et al. (2010),
and fieldwork sampling the water column under the sea ice of the McMurdo Sound (November 9
– 23, 2009), reported in Noble et al. (2013). Dissolved cobalt and pigment data from these three
fieldwork expeditions were sampled and analyzed with comparable methodologies as those used
on the CICLOPS expedition, and the CORSACS data are accessible online at https://www.bco-
dmo.org/dataset/3367.
2.8 Statistical Analysis
The linear regressions presented in this study are two-way (type-II) linear regressions, with
the exception of the standard addition curves used to calculate dCo concentrations (Sect. 2.2).
Two-way regressions are ideal for stoichiometric ratios because they allow for error in both the x
and y parameters and do not assume dependence between the x and y axes. The two-way regression
function used in this study was rewritten to Python from a MATLAB file (lsqfitma.m) originally
written by Ed Pelzer circa 1995 (Chmiel et al., 2022) and is available at https://github.com/rebecca-
chmiel/GP15.
Independent t-tests were performed using the stats.ttest_ind function within statistical
function module of the SciPy Python library.



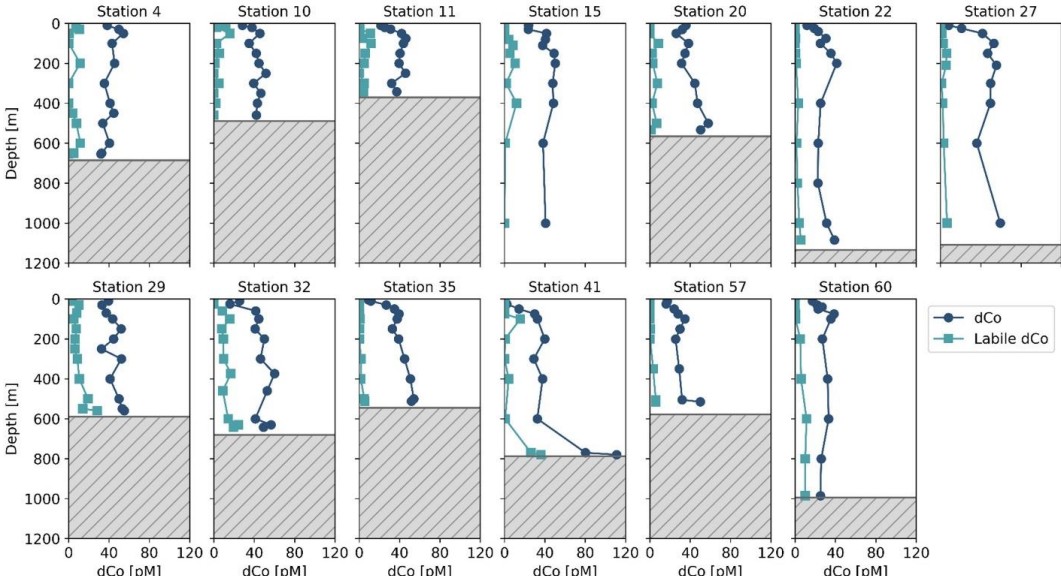

**Figure 2.** Dissolved Co and labile dCo full-depth profiles from the CICLOPS expedition to the
Amundsen Sea (Stations 4, 10, 11, 15), Ross Sea (Stations 20, 29, 32, 35) and Terra Nova Bay
(Stations 22, 27, 41, 57, 60). The top of the grey box marks the location of the seafloor.
**3 Results**
3.1 Dissolved Co distribution and speciation

During the CICLOPS expedition, full-depth profiles of dCo and labile dCo samples were
analyzed from 13 stations in the Amundsen Sea (Stations 4, 10, 11, 15), the Ross Sea (Stations 20,
29, 32, 35) and Terra Nova Bay (Stations 22, 27, 41, 57, 60; Fig. 1). The resulting dCo profiles
(Fig. 2) show depletion in the surface ocean consistent with a nutrient-type profile; at 10 m depth,
dCo concentrations were found to be $28 \pm 7$ pM in the Amundsen Sea (n = 4), $28 \pm 12$ pM in the
Ross Sea (n = 4), and only $11 \pm 7$ pM in Terra Nova Bay (n = 5; Table 1). Labile dCo distributions
generally followed those of dCo, and also showed strong depletion in the surface ocean. In the
Amundsen and Ross Seas, surface (~10 m) labile dCo concentrations ranged between 12 pM at
station 10 and undetected (n.d.) concentrations at stations 15, 20, 32 and 35. In Terra Nova Bay,
no surface labile dCo concentrations were detected at any of the 5 stations sampled, indicating that
the dCo inventory was dominated by the strongly ligand-bound dCo fraction.

In the deep ocean ($\geq 100$ m depth), dCo distributions were relatively consistent throughout
the water column, with the exception of elevated concentrations of dCo at near-bottom depths. The
Amundsen Sea, Ross Sea, and Terra Nova Bay all displayed similar deep ($\geq 100$ m depth) dCo
concentrations of $41 \pm 5$ pM (n = 30), $46 \pm 8$ pM (n = 32), and $39 \pm 18$ pM (n = 34), respectively
(Table 1). The high standard deviation of deep dCo in Terra Nova Bay is partially driven by the
elevated near-seafloor signal at Station 41; when the two deepest points at Station 41 are omitted
(770 m and 780 m), the average deep dCo in Terra Nova Bay was $36 \pm 10$ pM. The CICLOPS
expedition included regular near-bottom sampling as allowed by the altimeter aboard the trace
metal rosette. As a result, many of the deepest profile samples contained elevated concentrations



of dCo and labile dCo along the seafloor, including stations 20, 22, 27, 29, 32, 41 and 57. This deep dCo signal was particularly observable in stations where two near-seafloor samples were taken: one ~10 m above the seafloor and a second ~20 m above the seafloor. At stations 41 and 57, the elevated near-seafloor dCo signal was pronounced (Fig. 2); the samples ~10 m above the seafloor contained 111 pM and 50 pM dCo, respectively, which represents a 31 pM and 18 pM increase, respectively, from the samples collected ~20 m above the seafloor. This finding indicates that dCo was elevated in a narrow band close to the seafloor, and it is likely that dCo concentrations continued to increase in the 10 m between the deepest samples and the seafloor.

3.2 Phytoplankton communities in the Amundsen Sea, Ross Sea and Terra Nova Bay

Stations 11, 15, 22 and 27 exhibited high surface chlorophyll-a (Chl-a) fluorescence (17–42 mg m$^{-3}$ at 10 m), characteristic of phytoplankton blooms. The Amundsen Sea stations displayed high concentrations of 19'-hexanolyoxyfucoxanthin (19'-Hex), a pigment commonly used as a proxy for haptophyte biomass. In the coastal Southern Ocean, 19'-Hex is often correlated with *Phaeocystis antarctica* (DiTullio and Smith, 1996; DiTullio et al., 2003), and it is typical to find concentrated blooms of *P. antarctica* in these regions, particularly during the highly productive spring blooms of the Antarctic polynyas (Arrigo et al., 1999; DiTullio et al., 2000). The pigment fucoxanthin (Fuco) is commonly used as a proxy for diatom biomass, although it can also be produced by haptophytes like *P. antarctica* growing under Fe-replete conditions (DiTullio et al., 2003; DiTullio et al., 2007); Fuco was observed at stations throughout the expedition and tended to be relatively consistent throughout the CICLOPS stations, particularly in comparison to 19'-Hex, which displayed very high concentrations at some stations and much lower concentrations at others. In general, higher concentrations of Fuco were observed within Terra Nova Bay as well as at stations sampled later in the summer season. This is consistent with past observations of summer diatom blooms, which tend to occur after the annual spring bloom where and when dFe is available (Sedwick et al., 2000; Peloquin and Smith, 2007; Saito et al., 2010).

The upper ocean inventories of three pigments, 19'-Hex, Fuco and Chl-a, a proxy for general phytoplankton biomass in the Southern Ocean, were estimated via trapezoidal integration of their profiles between 5 and 50 m depth and compared to the 2005/2006 summer bloom observed on the CORSACS-1 expedition (Fig. 3). In the Ross Sea and Terra Nova Bay, CICLOPS stations contained smaller inventories of Chl-a and 19'-Hex compared to the Amundsen Sea, likely reflecting the end of the spring bloom and transition to a summer phytoplankton assemblage in these regions. One noticeable difference between the overlapping 2006 and 2018 January seasons is the larger Fuco inventory in 2006 in both the Ross Sea and Terra Nova Bay compared to the 2018 season, indicating a larger presence of diatom biomass during the CORSACS-1 expedition compared to the CICLOPS expedition despite relatively similar Chl-a inventories.

3.3 dZn, dCd and trace metal uptake rates

Dissolved Cd and Zn profiles, as well as trace metal uptake rate ($\rho$M) profiles for Co, Zn and Cd from the CICLOPS expedition were originally presented in Rao (2020) and Kellogg (2022). This study presents a comparison between dCo distribution and the distribution and uptake of dZn and dCd, two trace metals linked with Co biogeochemical cycling since all three metals are known to share similar uptake transporter pathways and can be interchangeably utilized as cofactors within specific classes of the enzyme carbonic anhydrase (Sunda and Huntsman, 1995, 2000; Saito and Goepfert, 2008; Kellogg et al., 2020, 2022).



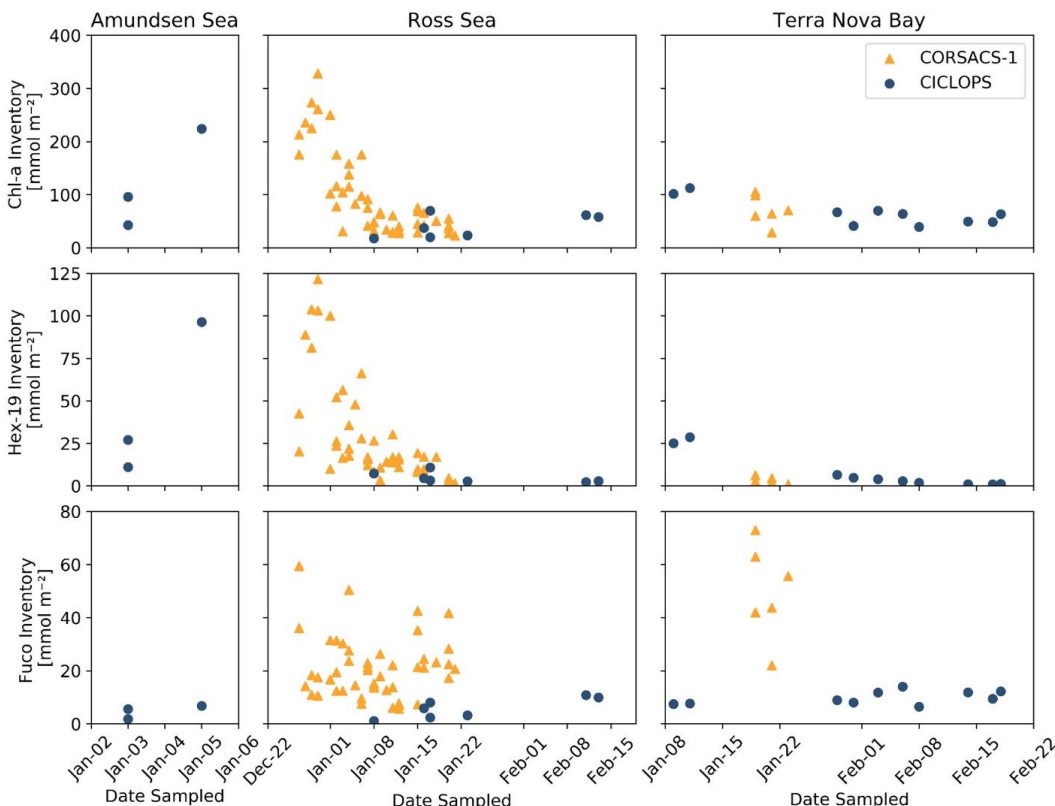

**Figure 3.** Upper ocean inventories of Chlorophyll-a (Chl-a), 19'-hexanolyoxyfucoxanthin (19'-
Hex) and fucoxanthin (Fuco) plotted over the austral summer season for both the 2005/2006
CORSACS-1 and 2017/2018 CICLOPS expeditions. Inventories were estimated via trapezoidal
integration of the pigment depth profiles between 5 and 50 m depth. Note that the dates along the
x-axis are not continuous between plots of each region, and the y-axis scales differ among the 3
pigments.
The dZn and dCd profiles observed on the CICLOPS expedition displayed nutrient-like
structure, with depleted concentrations near the surface (Fig. 4). In the deep ocean (≥100 m), dZn
and dCd concentrations were relatively uniform, displaying average deep concentrations of 4.6 ±
1.1 pM (n = 182) and 700 ± 90 pM, respectively (Table 2). Average dissolved metal concentrations
in the surface ocean (10 m depth) were higher in the Amundsen Sea (2.5 ± 1.2 nM dZn; 450 ± 170
pM Cd) compared to the Ross Sea (1.1 ± 1.2 nM dZn; 250 ± 170 pM dCd) and Terra Nova Bay
(0.87 ± 0.42 nM dZn; 130 ± 170 pM dCd). This trend of decreasing surface dissolved metals from
the Amundsen to Terra Nova Bay was mirrored in the dCo distributions, and could be explained
by the seasonal drawdown of metal nutrients in the mixed layer over time, differences in the metal
uptake of phytoplankton in the different regions, or both phenomenon occurring simultaneously.
At Stations 4, 11, 20, 22 and 57, uptake rates of Co, Zn and Cd within surface seawater
collected from 0–200 m were determined via spiked-isotope incubations (Rao, 2020; Kellogg,
2022). The relative ratios of the resulting uptake profiles from biomass collected onto 3 μm filters





provide insight into the demand for Co, Zn and Cd of eukaryotic phytoplankton in coastal
Antarctica (Fig. 5). Note that Co uptake within the bacterial size fraction (0.2–3 µm) was also
analyzed and the results are presented in Rao (2020), but here we present the results of the
eukaryotic size fraction (> 3 µm) to best represent the eukaryotic phytoplankton community
present and compare to the Zn and Cd uptake experiments. It should be noted that uptake rates
measured via tracer addition and shipboard incubations represent potential uptake and may be
overestimations of the environmental nutrient uptake rates because the isotope tracer addition was
labile – not at equilibrium with the natural seawater ligands – and could have perturbated the
natural micronutrient inventories. The $^{57}CoCl_2$ addition (0.1 pM) was likely a small enough
addition that the inventory was not significantly disturbed, but added concentrations of $^{67}ZnO$ (2
nM) and $^{110}CdO$ (300 pM) spikes were not tracer-level additions and necessarily increased the
existing trace metal inventories, possibly leading to the overestimation of total metal uptake rates
(Rao, 2020; Kellogg, 2022).
**Table 2.** Mean dZn and dCd values from the surface ocean (10 m) and the deep ocean (> 100 m)
in the three regions sampled.

| | Surface (10 m) | | | |
|---|---|---|---|---|
| **Region** | $dZn_{mean}$ [nM] | $n_{dZn}$ | $dCd_{mean}$ [pM] | $n_{dCd}$ |
| Amundsen Sea | 2.6 ± 1.2 | 4 | 450 ± 170 | 4 |
| Ross Sea | 1.1 ± 1.2 | 6 | 250 ± 170 | 7 |
| Terra Nova Bay | 0.87 ± 0.42 | 11 | 130 ± 60 | 11 |
| All | 1.3 ± 1.0 | 21 | 230 ± 170 | 22 |
| | Deep (> 100 m) | | | |
| **Region** | $dZn_{mean}$ [nM] | $n_{dZn}$ | $dCd_{mean}$ [pM] | $n_{dCd}$ |
| Amundsen Sea | 5.4 ± 0.6 | 30 | 730 ± 40 | 30 |
| Ross Sea | 4.7 ± 0.6 | 65 | 740 ± 80 | 65 |
| Terra Nova Bay | 4.3 ± 1.4 | 87 | 670 ± 100 | 90 |
| All | 4.6 ± 1.1 | 182 | 700 ± 90 | 185 |


Of the five stations with uptake rate data from all three trace metals of interest, four
(Stations 4, 11, 20 and 22) were from a transect conducted from the Amundsen Sea to Terra Nova
Bay, and were sampled within a span of 10 days from December 31, 2017 to January 9, 2018,
while the last station (Station 57) was sampled later in the summer on February 6, 2018; this range
of stations allows us to assess the uptake stoichiometry along both spatial (location) and time
(bloom progression) dimensions. The $\rho M$ profiles displayed an increase in metal uptake of Co, Zn
and Cd towards the surface, a shape which was mirrored in the lower dissolved trace metal
concentrations of the surface ocean, suggesting the influence of phytoplankton uptake on the
drawdown of micronutrients in the photic zone. The stoichiometry of $\rho M$ among Co, Zn and Cd
tended to directly follow the metals' availability as dissolved species: Co, which is present at the
lowest concentrations of ~$10^{-11}$ M, was taken up at rates ranging between $10^{-13}$ and $10^{-12}$ M d$^{-1}$;
Cd, at concentrations of ~$10^{-10}$ M, was taken up at rates of $10^{-12}$ to $10^{-11}$ M d$^{-1}$; and Zn, present in
the highest concentration of ~$10^{-9}$ M, was taken up at rates of $10^{-12}$ to $10^{-10}$ M d$^{-1}$. This observation





reveals order-of-magnitude differences in biological uptake between the three metals, matching
patterns of metal availability in the water column.

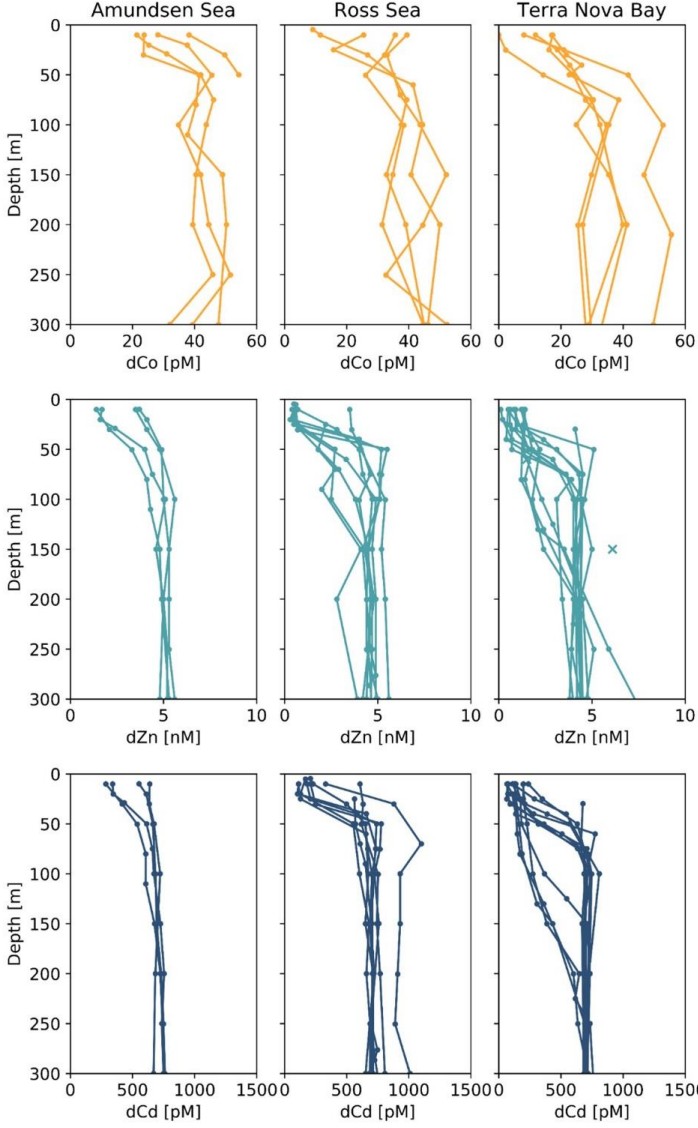

**Figure 4.** Upper ocean trace metal depth profiles of dCo, dZn and dCd, by region (left panels,
Amundsen Sea; middle panels, Ross Sea; right panels, Terra Nova Bay). Outliers are marked with
an 'x'. Dissolved Zn and Cd profile data are further described in Kellogg (2022).



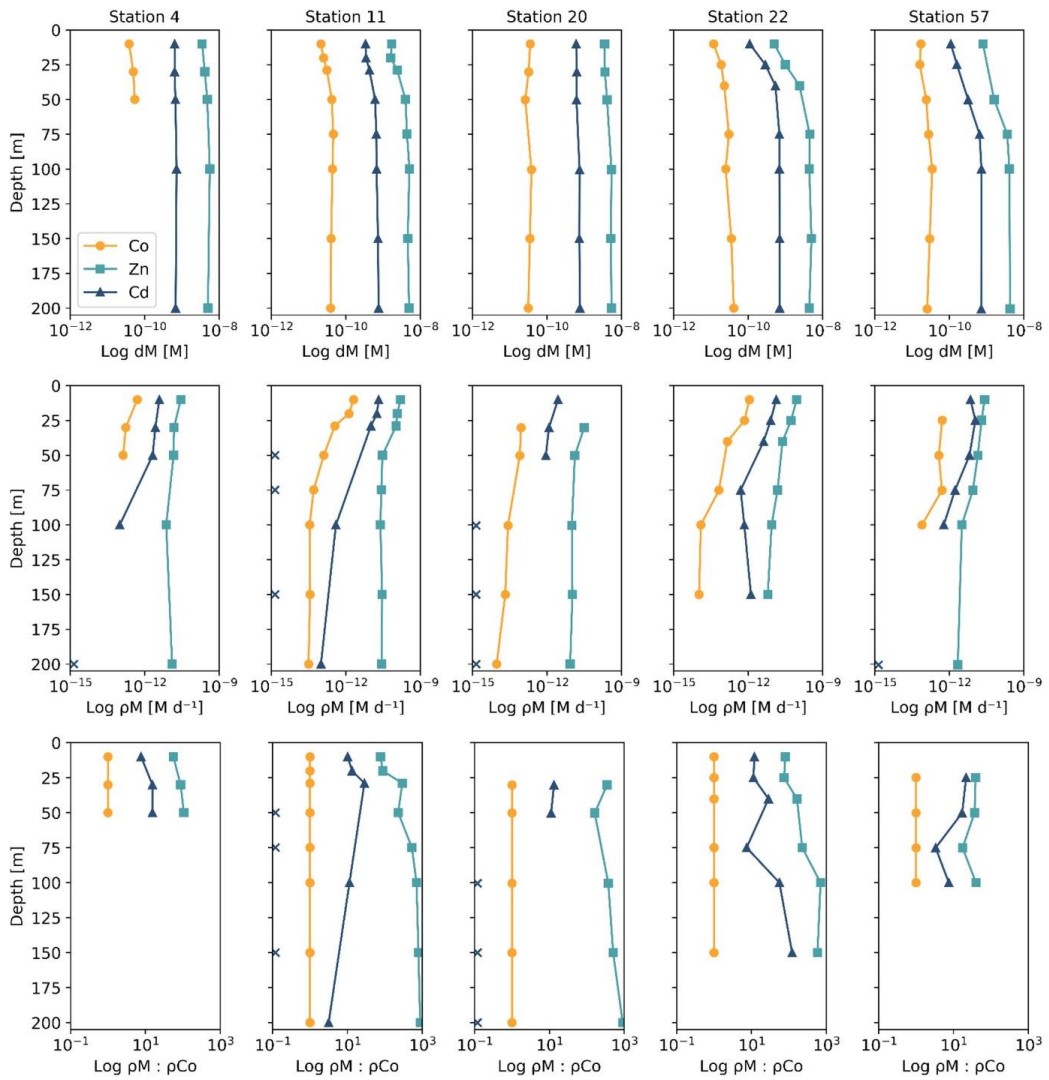

**Figure 5.** Depth profiles of dissolved metals (dM; top), trace metal uptake rates (ρM; middle), and trace metal uptake rates normalized to the uptake rate of dCo (ρM : ρCo), plotted along a log scale. Stations 4 and 11 are from the Amundsen Sea, Station 20 is from the Ross Sea, and Stations 22 and 57 are from Terra Nova Bay. Depths at which an uptake rate below detection (specifically for ρCd) are marked with an 'x' along the y-axis. Co trace metal uptake data are further described in Rao (2020) and Zn and Cd uptake data are further described in Kellogg (2022).

## 4 Discussion

4.1 Biogeochemical Co cycle processes observed via dCo profiles and dCo : dPO$_4^{3-}$ stoichiometry

Low surface ocean dCo and labile dCo concentrations are attributable to uptake by phytoplankton and bacteria in the Southern Ocean, giving the dCo and labile dCo profiles a distinct



nutrient-like shape (Fig. 2). The labile dCo fraction was extremely low or below the limit of detection in surface waters, particularly within Terra Nova Bay, indicating strong drawdown of the labile fraction and near 100% complexation of dCo in the water column. Labile dCo is considered to be more bioavailable than strongly-bound dCo and thus is likely preferentially taken up by microbes when available. This labile dCo may then be rapidly cycled by phytoplankton in the mixed layer and any labile dCo released via remineralization, cell lysis, or grazing would be promptly taken up by other algae and microbes. A rapid turnover of labile dCo suggests a high demand for bioavailable Co from the surface phytoplankton community.

Dissolved Co and $PO_4$ displayed a generally positive relationship in the upper ocean, which is indicative of the co-cycling of both nutrients via phytoplankton uptake and remineralization (Fig. 6a). The processes of biological uptake and remineralization, when observed along dCo vs. $dPO_4^{3-}$ axes, can be represented by vectors with positive slopes and opposite directionality. Abiotic dCo inputs and Co scavenging processes can be represented by vertical or near-vertical vectors because they decouple the cycling of dCo and $dPO_4^{3-}$. The positive dCo vs. $dPO_4^{3-}$ linear relationship that is often observed within the ocean's mixed layer can exhibit a variety of slopes that are dictated by the nutrient uptake and remineralization stoichiometry of the microbial community (Saito et al., 2017). On CICLOPS, the dCo vs. $dPO_4^{3-}$ relationship displayed a drawdown of both dCo and $dPO_4^{3-}$ in the upper ocean, and the labile dCo vs. $dPO_4^{3-}$ relationship revealed the stark lack of labile dCo throughout the upper ocean (Fig. 6b,d). The dCo vs. $dPO_4^{3-}$ slope in the upper ocean (0–100 m depth) was found to be distinct for each of the three regions sampled on the expedition; the Ross Sea displayed the highest slope ($74 \pm 18$ µmol : mol), followed by the Amundsen Sea ($47 \pm 9$ µmol : mol) and Terra Nova Bay, which displayed the lowest dCo vs. $dPO_4^{3-}$ slope ($26 \pm 4$ µmol : mol; Fig. 7; Table 3). These slopes reflect a relatively wide range of dCo stoichiometries that vary by a factor of 2.8 between the lowest and highest slopes observed. For comparison, the 2005/2006 CORSACS-1 and CORSACS-2 Ross Sea data points were pooled and the dCo vs. $dPO_4$ slope was recalculated (originally reported as 37.6 µmol : mol between 5–500 m depth; Saito et al., 2010) to fall within the same depth window (0–100 m). The resulting slope fell within the range of slopes observed on CICLOPS ($49 \pm 4$ µmol : mol; $R^2 = 0.57$; n = 106).

The range of dCo vs. $dPO_4^{3-}$ slopes reflects the elasticity of cobalt uptake stoichiometry in the upper ocean, which varies by microbial community and the availability of dCo and other nutrients. Due to the number of factors that can affect the environmental stoichiometry of trace metal nutrients, the dCo vs. $dPO_4^{3-}$ slope must be interpreted alongside other information about the marine environment, such as the available dCo inventory and the local nutrient limitation regime, making global comparisons of dCo : $dPO_4^{3-}$ stoichiometry complex. The lower stoichiometric slope observed in Terra Nova Bay compared to the Ross and Amundsen Seas likely indicates not a lack of demand for Co by phytoplankton, but the low availability of Co in the surface ocean despite high demand for the metal. Terra Nova Bay was found to have the lowest average surface dCo, dZn and dCd concentrations of the three regions studied, and both Terra Nova Bay stations where $\rho$Co was measured (Stations 22 and 57) displayed higher surface Co uptake rates (0.71 and 0.51 pM d$^{-1}$, respectively, at 25 m depth) than Station 20 in the Ross Sea (0.09 pM d$^{-1}$ at 30 m depth). It is likely that the lower dCo stoichiometry in Terra Nova Bay was driven by nutrient draw-down and low availability of labile dCo in the region resulting from productive phytoplankton blooms. Remineralization would also have played a role in setting the dCo vs. $dPO_4$ slope below the photic zone; a remineralization vector with a relatively low slope indicates that there was a lower dCo source from particulate Co biomass and a rapid turnover of





recycled dCo back into biomass, suggesting a tight coupling of the dissolved and particulate
phases.

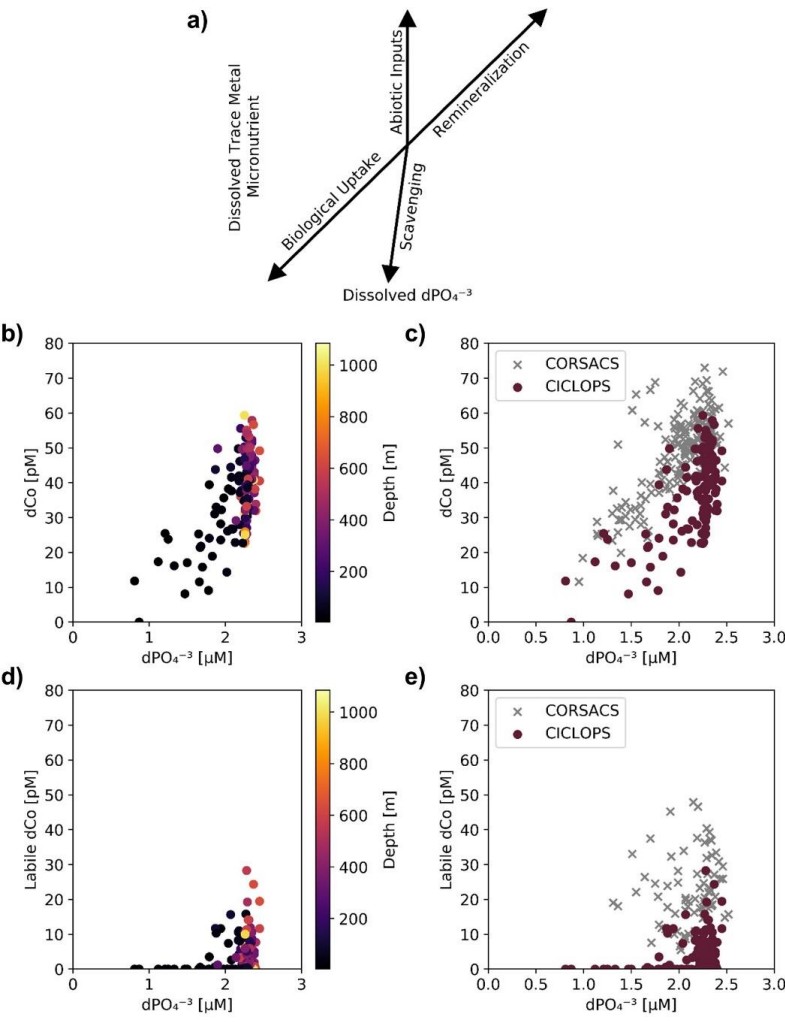

**Figure 6. (a)** A vector schematic of the relationship between $dPO_4^{3-}$ and dissolved trace metals
like dCo, and how the various marine processes can affect their distribution and environmental
stoichiometry. Adapted from Noble et al. (2008). The CICLOPS **(b)** dCo vs. $dPO_4^{3-}$ relationship
and **(d)** labile dCo vs. $dPO_4^{3-}$ relationship, plotted by depth. Also shown are the CICLOPS (red)
**(c)** dCo vs. $dPO_4^{3-}$ and **(e)** labile dCo vs. $dPO_4^{3-}$ samples overlaid with CORSACS (gray) samples.

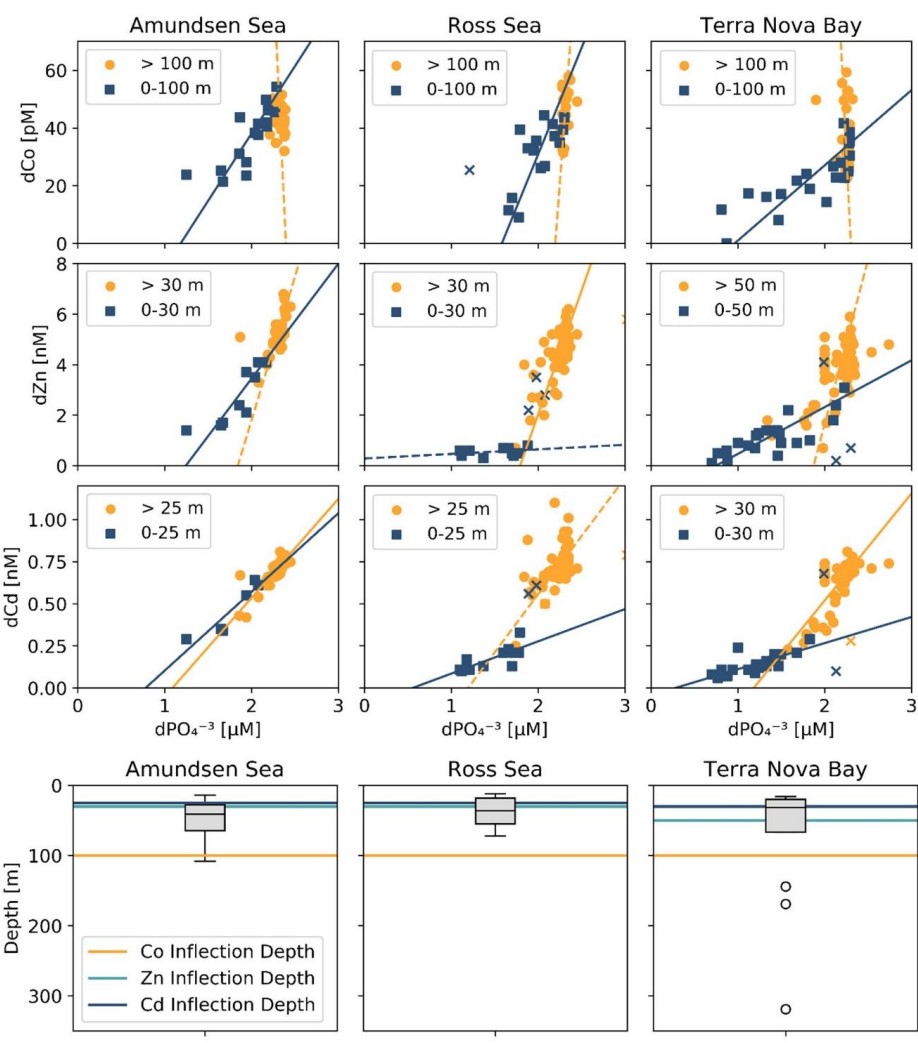

**Figure 7.** (Top 3 rows) Trace metal : $dPO_4^{3-}$ relationships from the three CICLOPS regions sampled, divided into upper ocean (blue square) and deep ocean (orange circle) bins with a manual depth threshold (or inflection point depth) selected to optimize the linear fit of the upper and deep ocean trends. Regressions with an $R^2 \geq 0.50$ are shown as a solid line, and those with an $R^2 < 0.50$ are shown as a dotted line. The results of the linear regressions are given in Table 3. Regression outliers are marked with an 'x'. (Bottom row) The inflection point depths assigned to dCo, dZn and dCd relationships are shown compared to a box and whiskers plot of the mixed layer depths, with mixed layer depth outliers marked with an 'o'.





**Table 3.** Trace metal : dPO$_4^{3-}$ stoichiometric regressions for dCo, dZn and dCd in both the surface and deep ocean of the Amundsen Sea, Ross Sea and Terra Nova Bay, as shown in Fig. 7. Linear regression slopes with $R^2 < 0.50$ are not shown as the slope values should not be considered meaningful stoichiometric values.

| Region | dCo:dPO$_4^{3-}$ [μmol:mol] | | | | dZn:dPO$_4^{3-}$ [mmol:mol] | | | | dCd:dPO$_4^{3-}$ [mmol:mol] | | | |
|---|---|---|---|---|---|---|---|---|---|---|---|---|
| | Depths [m] | n | Slope | $R^2$ | Depths [m] | n | Slope | $R^2$ | Depths [m] | n | Slope | $R^2$ |
| **Amundsen Sea** | | | | | | | | | | | | |
| Surface | 0-100 | 16 | 47 ± 9 | 0.64 | 0-30 | 9 | 4.6 ± 0.9 | 0.72 | 0-25 | 6 | 0.47 ± 0.08 | 0.86 |
| Deep | >100 | 20 | -- | 0.02 | >30 | 35 | -- | 0.37 | >25 | 38 | 0.59 ± 0.06 | 0.72 |
| **Ross Sea** | | | | | | | | | | | | |
| Surface | 0-100 | 15 | 74 ± 18 | 0.53 | 0-30 | 11 | -- | 0.07 | 0-25 | 11 | 0.19 ± 0.05 | 0.56 |
| Deep | >100 | 24 | -- | 0.21 | >30 | 77 | 9.8 ± 1.0 | 0.54 | >25 | 79 | -- | 0.26 |
| **Terra Nova Bay** | | | | | | | | | | | | |
| Surf | 0-100 | 20 | 26 ± 4 | 0.65 | 0-50 | 24 | 1.9 ± 0.3 | 0.65 | 0-30 | 21 | 0.15 ± 0.03 | 0.59 |
| Deep | >100 | 26 | -- | 0.05 | >50 | 95 | -- | 0.30 | >30 | 104 | 0.64 ± 0.03 | 0.80 |

Deviations from the linear uptake-remineralization line in the dCo vs. dPO$_4^{3-}$ relationship occur when dCo distributions become decoupled from dPO$_4^{3-}$ or vice versa, as with Co scavenging onto particles and lithogenic dCo sources. In other ocean regions, the dCo vs. dPO$_4^{3-}$ relationship displays a characteristic "curl" towards the high- dPO$_4^{3-}$, low-dCo in deeper waters, resulting from the net vector sum of both remineralization, which increases both dPO$_4^{3-}$ and dCo, and scavenging to Mn-oxides, which removes dCo in excess of dPO$_4^{3-}$ from the water column (Noble et al., 2008; Hawco et al., 2017; Saito et al., 2017). The dCo vs. dPO$_4^{3-}$ relationship observed on CICLOPS, however, displayed no such scavenging curl, indicating no clear signal of dCo loss due to scavenging, at least within timescales relevant to water column mixing. This finding is consistent with previous studies of the Ross Sea that have also observed little evidence of dCo loss via scavenging in the mesopelagic (Saito et al., 2010; Noble et al., 2013). The lack of a visible scavenging signal may be attributable to the deep winter mixed layers of coastal Antarctic seas that reach depths of up to 600 m and can extend to the seafloor (Smith and Jones 2015). This deep vertical mixing allows the dCo : dPO$_4^{3-}$ ratio in the deep ocean to reset on an annual timescale, potentially erasing any signals of dCo scavenging, which would be expected to occur on a timescale of decades to centuries (Hawco et al., 2017). Additionally, Oldham et al. (2021) concluded that a suppressed Co scavenging flux might be the result of a unique Mn cycle in the Ross Sea, characterized by low to undetectable concentrations of Mn-oxide particles, slow rates of Mn-oxide formation, and the stabilization of organic dMn via Mn(III) ligands (Oldham et al., 2021).

The elevated dCo signal observed from several depths within 20 m of the seafloor were sourced from a benthic nepheloid layer: a near-seafloor region of the water column characterized by high particle abundance, turbulence, and isopycnal movement of both dissolved and particulate material along the seafloor (Gardner et al., 2018). The Ross Sea has been observed to display strong nepheloid layers as cold, dense water flows northward along the Ross Sea shelf until it reaches the shelf break, carrying suspended sediments with it along the seafloor (Budillon et al., 2006). Nepheloid layers tend to be enriched in dissolved trace metals like dFe, and can act as a source of micronutrients if upwelled to the surface ocean (Marsay et al., 2014; Noble et al., 2017). Elevated dCo concentrations within the Ross Sea nepheloid layer is a novel finding, as previous expeditions analyzing dCo concentrations in the Ross Sea did not sample as close to the seafloor





as the CICLOPS trace metal rosette was able to (Fitzwater et al., 2000; Saito et al., 2010; Noble et
al., 2013). This finding is evidence of a dCo source to the deep ocean that may be upwelled to
intermediate and upper ocean waters via vertical mixing.
4.2 Decreased Ross Sea dCo and labile dCo inventories
The dCo and labile dCo profiles observed along the 2017/2018 CICLOPS expedition
displayed similar vertical structure as those observed along the 2005/2006 CORSACS expeditions;
however, the CICLOPS dCo and labile dCo concentrations were notably lower throughout the
water column compared to the CORSACS datasets (Fig. 8). This trend was particularly clear in
the Ross Sea, where the stations from both expeditions contained the greatest regional overlap and
labile dCo distributions from the prior 2006 CORSACS-2 expedition exceeded those observed on
the 2017/2018 CICLOPS expedition (Fig. 9a-c; Table 4). The CORSACS-1 and CORSACS-2
expeditions displayed average deep ($\geq$ 100 m) dCo concentrations of $55 \pm 4$ pM and $56 \pm 6$ pM,
respectively, and CORSACS-2 displayed average deep labile dCo concentrations of $21 \pm 7$ pM;
on CICLOPS, in contrast, the Ross Sea displayed average deep dCo and labile dCo concentrations
of $46 \pm 8$ pM and $9 \pm 7$ pM, respectively. Independent t-tests determined that CORSACS-1 and
CORSACS-2 deep Ross Sea dCo values were statistically similar ($p = 0.27$) while deep CICLOPS
dCo values were statistically different from CORSACS-1 and CORSACS-2 deep dCo ($p < 0.0001$;
Table 4). This offset represents a mean dCo inventory loss of 8 – 10 pM dCo in the deep ocean,
and approximately all of the difference can be accounted for by the loss of deep labile dCo (12 pM
dCo; Fig. 9d-g), the more bioavailable form of dCo for biological uptake.
**Table 4.** The mean dCo and labile dCo observed in the deep ($\geq$ 100 m) Ross Sea, and the average
deep dCo loss between 3 previous sampling expeditions (CORSACS-1 in summer 2005/2006;
CORSACS-2 in spring 2006; under-ice sampling in McMurdo Sound in spring 2009) and the
CICLOPS expedition (2017/2018). Dissolved Co and labile dCo loss values were calculated as the
difference between mean deep concentrations observed on previous expeditions and those
observed on the CICLOPS expedition. No labile dCo data (n.d.) is presented from the CORSACS-
1 expedition. Independent t-tests were performed to determine the significance of difference
between the deep mean concentrations from previous expeditions compared to the CICLOPS
expedition; * indicates a significant difference between CICLOPS and a previous expedition ($p <$
0.005). The mean deep dCo concentrations from the CORSACS expeditions were not significantly
different from each other ($p = 0.27$).

| | $dCo_{mean}$ [pM] | n | Labile $dCo_{mean}$ [pM] | n | dCo Loss [pM] | *p*-value | Labile dCo Loss [pM] | *p*-value |
|---|---|---|---|---|---|---|---|---|
| CORSACS-1[a] | $55 \pm 4$ | 26 | n.d. | | $8 \pm 9$ | $< 0.0001*$ | -- | -- |
| CORSACS-2[a] | $56 \pm 6$ | 19 | $21 \pm 7$ | 20 | $10 \pm 10$ | $< 0.0001*$ | $12 \pm 10$ | $< 0.0001*$ |
| McMurdo Sound[b] | $51 \pm 4$ | 19 | $15 \pm 2$ | 19 | $4 \pm 8$ | 0.02 | $6 \pm 7$ | $0.0006*$ |
| CICLOPS | $46 \pm 8$ | 32 | $9 \pm 7$ | 32 | -- | -- | -- | -- |

[a] Data originally published in Saito et al. (2010).
[b] Data originally published in Noble et al. (2013).

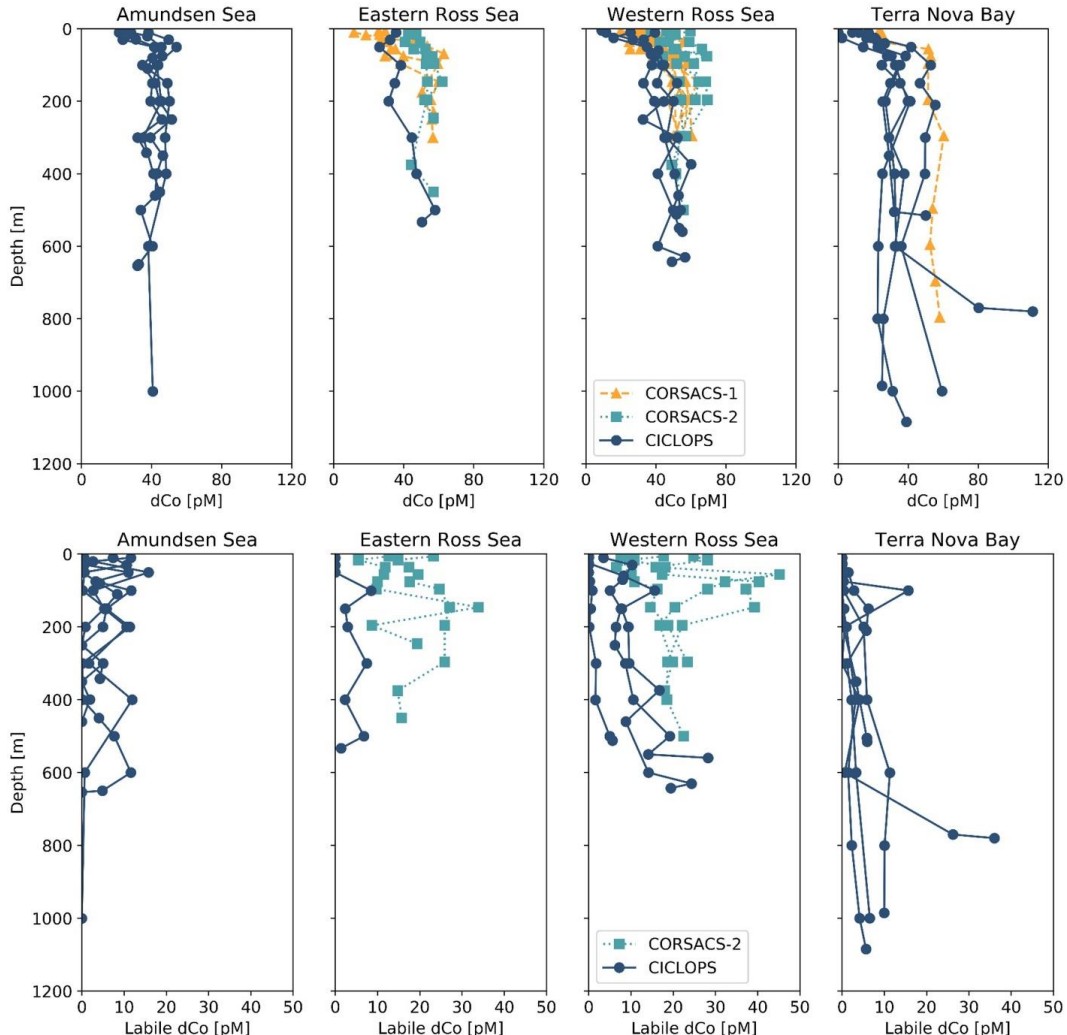

**Figure 8.** Dissolved Co and labile dCo depth profiles from the CORSACS-1 (NBP0601; December 27, 2005 – January 23, 2006), CORSACS-2 (NBP0608; November 8, 2006 – December 3, 2006) and CICLOPS (NBP-1801; December 11, 2017 – March 3, 2018) expeditions in the 4 regions sampled by the CICLOPS expedition: Terra Nova Bay, the Western Ross Sea, the Eastern Ross Sea and the Amundsen Sea. The Eastern and Western Ross Sea stations are defined by being either east or west of the 175 ºE longitudinal, respectively. The CORSACS expeditions did not extend to the Amundsen Sea, and no labile dCo was reported from the CORSACS-1 expedition. dCo data from the CORSACS expeditions was reported in Saito et al. (2010) and is accessible at https://www.bco-dmo.org/dataset/3367.



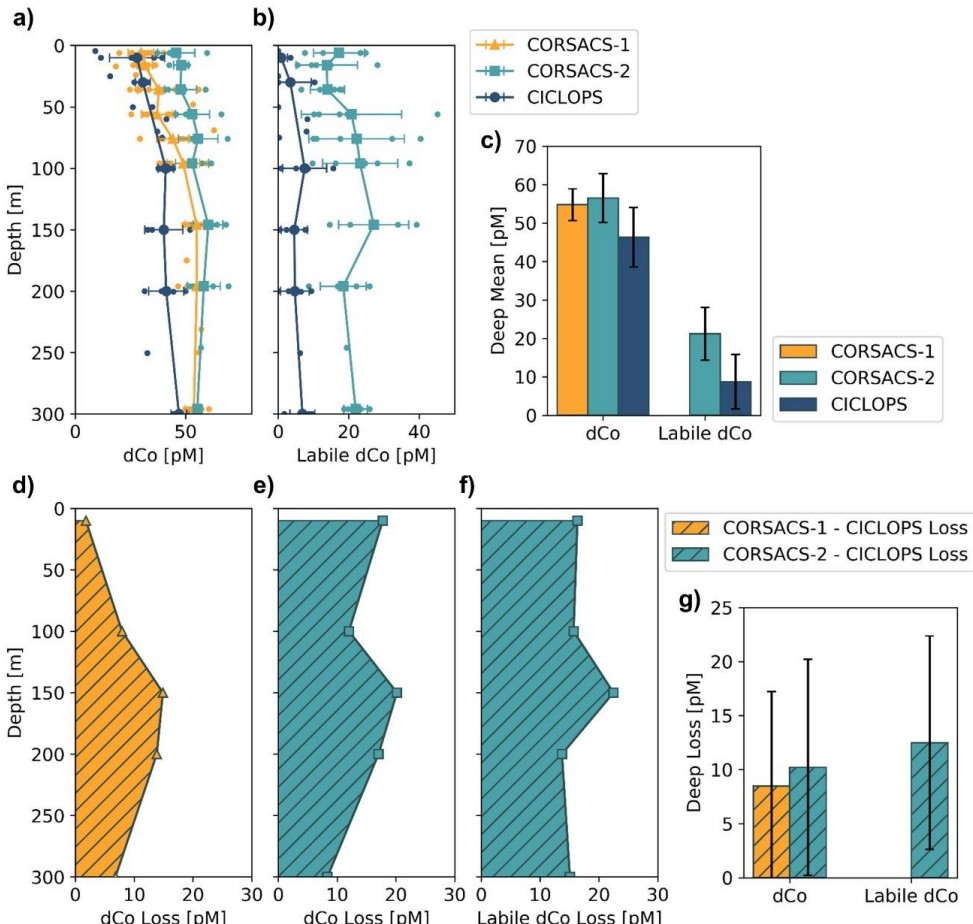

**Figure 9.** Mean depth profiles of dCo **(a)** and labile dCo **(b)** from the Ross Sea from three sampling
seasons, including the expeditions: CORSACS-1 (Summer 2005/2006), CORSACS-2 (Spring
2006) and CICLOPS (Summer 2017/2018). Observed profile values are plotted as unconnected
dots, and the mean profile is plotted for each depth at which at least three samples were analyzed.
**(c)** The mean deep ($\geq 100$ m) dCo and labile dCo concentrations for stations in the Ross Sea on
each expedition. The mean difference in the dCo **(d, e)** and labile dCo **(f)** profiles between the
CORSACS and CICLOPS expeditions where sample depths were within 5 m of each other. **(g)**
The mean deep ($\geq 100$ m) dCo and labile dCo concentration loss for stations in the Ross Sea. Error
bars denote one standard deviation from the mean. No labile dCo data is available for the
CORSACS-1 expedition. Mean values, loss values, and the results of independent t-tests to
determine the significance of the deep dCo loss are given in Table 4.
In the near-surface (10 m), labile dCo was undetectable at 3 of the 4 stations in the Ross
Sea on CICLOPS, and the near-surface labile : total dCo ratio in the one station where labile dCo
was detectable (station 29; 3.5 pM labile dCo) was only 0.09. In contrast, the 2006 CORSACS-2
expedition reported the presence of labile dCo at five stations with concentrations of $17 \pm 7$ pM at



6 m depth and $14 \pm 9$ pM at 16 m depth, and reported labile : total dCo ratios at 6 m and 16 m depth of $0.37 \pm 0.13$ and $0.28 \pm 0.17$, respectively. This trend can be at least partially explained by the seasonality differences between the spring CORSACS-2 expedition and the summer CICLOPS expedition; as the phytoplankton bloom progresses in the photic zone of the Ross Sea, labile dCo concentrations would be drawn down by community uptake and would exhibit lower concentrations later in the summer season. This seasonal trend was evident in the surface dCo inventory differences between the summer CORSACS-1 and spring CORSACS-2 expeditions (Fig. 9a,d,e). However, the low, often undetectable, labile dCo concentrations observed in the surface Ross Sea on the CICLOPS expedition illustrate the intensity of bloom-driven labile dCo depletion in the region, leaving 91–100% strong ligand-bound dCo in the surface Ross Sea. These observations are consistent with the Co uptake rate measurements, which were found to be higher on CICLOPS ($0.84$ pM d$^{-1}$, n = 38) compared to CORSACS-1 and CORSACS-2 ($0.67$ pM d$^{-1}$ and $0.25$ pM d$^{-1}$, respectively) (Saito et al., 2010; Rao, 2020).

Dissolved Co and labile dCo concentrations were also analyzed in the Ross Sea in 2009 by sampling the water column below the McMurdo Sound seasonal sea ice in the early spring (November 9–23) (Noble et al., 2013). Under the ice, the water column was well-mixed, and the dCo and labile dCo profiles showed relative uniformity at all three stations measured (Fig. 2 of Noble et al.,, 2013). In the deep ocean ($\geq 100$ m), the mean dCo and labile dCo concentrations were $51 \pm 4$ and $15 \pm 2$ pM, respectively, which is lower than those observed on the 2005/2006 CORSACS expeditions and higher than those observed on the 2017/2018 CICLOPS expedition (Table 4). The mean deep labile dCo concentrations from the McMurdo Sound fieldwork were also significantly different from the mean deep labile dCo observed on CICLOPS ($p = 0.0006$), displaying an average deep labile dCo difference of 6 pM. This dataset supports the possibility of a long-term trend towards a decreasing deep dCo inventory in the Ross Sea, although the more coastal location and difference in sea ice cover should be considered when comparing the McMurdo Sound dataset to the CORSACS and CICLOPS observations. Notably, the methodology and instrumentation used to measure both dCo and labile dCo on both CORSACS expeditions, the McMurdo Sound fieldwork and the CICLOPS expedition were functionally identical, with the exception of an autosampler (Metrohm 858 Sample Processor) used on the 2017/2018 CICLOPS expedition.

The low labile dCo inventory in the Ross Sea was a surprising discovery during CICLOPS since relatively high concentrations of labile dCo were previously noted to be a distinctive feature of the Ross Sea and Southern Ocean when compared to the tropical and subtropical global oceans (Saito et al., 2010). In prior studies in this region, high labile : total dCo ratios were hypothesized to be due to the absence of ligand-producing – and vitamin B$_{12}$-producing – marine cyanobacteria like *Synechococcus* in the Ross Sea (Caron et al., 2000; DiTullio et al., 2003; Bertrand et al., 2007), since *Synechococcus*-dominated communities have been known to produce a substantial amount of Co ligands (Saito et al., 2005). However, high Co ligand concentrations and low labile dCo concentrations have previously been observed at a more pelagic location in the Southern Ocean near New Zealand, where it was hypothesized that the decay of a eukaryotic phytoplankton bloom generated higher abundances of Co-binding ligands in the surface ocean (Ellwood et al., 2005).

The decrease in the dCo and labile dCo inventories was apparent when the CICLOPS and CORSACS dCo vs. dPO$_4^{3-}$ relationships across all expedition regions were compared (Fig. 6c,e). Over similar dPO$_4^{3-}$ ranges, the CICLOPS dCo concentrations are generally lower than those observed on CORSACS, and the CICLOPS labile dCo concentrations are considerably lower, with





labile dCo essentially absent from upper ocean samples with a dPO$_4^{3-}$ concentration < 1.75 μM. Despite the lack of observable scavenging, the CICLOPS dCo vs. dPO$_4^{3-}$ relationship appeared to be noticeably nonlinear throughout the water column ($R^2$ = 0.42), while CORSACS samples displayed a more linear trend ($R^2$ = 0.57). The CICLOPS dCo vs. dPO$_4^{3-}$ relationship creates a concave, "scooped" shape where dCo was depleted relative to dPO$_4^{3-}$, displaying a lower slope in the upper ocean than was observed on the CORSACS expeditions (Fig. 6c). This scooped shape was particularly evident in Terra Nova Bay where the upper ocean dCo : dPO$_4^{3-}$ stoichiometric slope was the lowest (26 ± 4 μmol : mol; $R^2$ = 0.65). The depletion of dCo relative to dPO$_4^{3-}$ observed on CICLOPS appears driven by the shift in Co speciation as a result of near-total uptake of the upper ocean labile dCo fraction and subsequent dominance of the remaining strong ligand-bound dCo fraction in the upper ocean. Similar to the deep dCo loss described above, the difference between the CORSACS and CICLOPS dCo vs. dPO$_4^{3-}$ relationship can be accounted for by the depletion of the labile dCo inventory. In the deep ocean where both dCo and dPO$_4^{3-}$ are more abundant, the large range in dCo concentrations relative to dPO$_4^{3-}$ concentrations may be evidence of deep inputs of dCo and labile dCo from the nepheloid layer, which was more attentively sampled on CICLOPS than either CORSACS expedition (Sect. 4.1).

4.3 Dissolved Co, Zn and Cd stoichiometry

Dissolved Zn concentrations observed on CICLOPS were low in the surface ocean, particularly in Terra Nova Bay, where dZn concentrations in the sub-nanomolar ranges were observed (average dZn = 0.87 ± 0.42 at 10 m depth, n = 11). Marine microbes require Zn for a wide range of metabolic uses; for example, eukaryotic phytoplankton use Zn as a cofactor in carbonic anhydrase (Roberts et al., 1997; Morel et al., 2020) and bacteria such as *Pseudoalteromonas* use Zn in a range of proteases (Mazzotta et al., 2021). Prior culture studies have found that Zn scarcity can lead to co-limitation of both Zn and carbon in several eukaryotic phytoplankton strains (Morel et al., 1994; Sunda and Huntsman, 2000), and field incubation experiments have shown evidence for Zn co-limitation with Fe (Jakuba et al., 2012) and silicate (Chappell et al., 2016) in the Pacific Ocean. During the CICLOPS expedition, an incubation experiment performed at Station 27 in Terra Nova Bay found compelling evidence for Zn and Fe co-limitation, which constrained Chl-a production and DIC draw-down by phytoplankton in the region (Kellogg et al., [Submitted]).

Many but not all phytoplankton are able to substitute Co and Cd for Zn as their carbonic anhydrase metallic cofactor (Lee and Morel, 1995; Sunda and Huntsman, 1995; Lane et al., 2005; Kellogg et al., 2022), which provides metabolic flexibility and a competitive edge in low-dZn environments (Kellogg et al., 2020). The Cd-containing carbonic anhydrase CDCA is currently the only known metabolic use of Cd, and the uptake of dCd and dCo in the photic zone, both metals which are typically less abundant than dZn in the oceans, often increases under low dZn conditions (Sunda and Huntsman, 1995, 1996; Jakuba et al., 2008; Kellogg et al., 2020; Morel et al., 2020). Cations like Zn, Cd and Co that possess similar charge and atomic radii often share the same transporter uptake systems, and the relative availability of different metal cofactors for use in an organism's metalloproteome is partially determined by the environmental metal concentrations and the affinity of the metals for ligands associated with a cell's metal transport proteins (Irving and Williams, 1948; Sunda and Huntsman, 1992, 1995). When dZn concentrations are low, more Cd and Co are able to bind to the transporter ligands despite the relative stability of their ligand-bound complexes, which tend to be lower for Co than for Zn. Through this mechanism, dZn





concentrations and cycling can influence the distribution and uptake of Co and Cd, particularly in low dZn environments like the Ross Sea and Terra Nova Bay.

The dZn vs. $dPO_4^{3-}$ and dCd vs. $dPO_4^{3-}$ relationships observed in the Amundsen Sea, Ross Sea and Terra Nova Bay were compared relative to dCo vs. $dPO_4^{3-}$ (Fig. 7; Table 3). The resulting shapes of these relationships were similar to that of dCo vs. $dPO_4^{3-}$, exhibiting distinct differences in slope between surface and deep waters. The stark difference in trace metal stoichiometry slopes between the upper and deep ocean is likely driven by differences in metal speciation over depth. In the surface ocean, a shallower trace metal : $dPO_4^{3-}$ slope suggests a trace metal fraction that is largely bound to strong organic ligands, with a smaller excess labile fraction. The more bioavailable labile fraction of metals would have been drawn down by phytoplankton, whose uptake transport systems preferentially bind to labile metals. At deeper depths, the presence of labile metals in excess of strong organic ligands results in a higher metal : $dPO_4^{3-}$ slope. For this analysis, the depth threshold that separates the upper ocean from the deep ocean was selected manually in order to optimize the linear fit of the upper and deep ocean trends and to best capture the depth dependence of the observed trace metal stoichiometries. This depth threshold can best be conceptualized as an inflection point that represents the largest change in trace metal concentrations with respect to depth or, in this case, $dPO_4^{3-}$ concentration. The depth threshold used for dCo in both the Ross Sea and Terra Nova Bay (100 m) is deeper than those used for dZn and dCd, (range of 25 – 50 m). Thus, the inflection points of the "scoops" in the trace metal stoichiometries are driven by the uptake stoichiometry of the region's phytoplankton community rather than the mixed layer depth of the upper ocean.

A shallow dCo : $dPO_4^{3-}$ slope that extends below the photic zone could suggest Co uptake by heterotrophic bacteria, archaea and possibly sinking phytoplankton below the photic zone. Heterotrophic prokaryotic uptake of labile Co is largely driven by the bacteria and archaea that contain a vitamin $B_{12}$ synthesis pathway that is absent in all eukaryotes (Warren et al., 2002; Osman et al., 2021); unlike carbonic anhydrase, the use of Co as a co-factor in the vitamin $B_{12}$ corrin ring structure cannot be substituted for by other divalent cations like Zn and Cd. Many vitamin $B_{12}$-synthesizing bacteria possess genes for Co(II)-specific transporters in addition to more general metal ion transporters, and the Co-specific transporters are regulated by cellular concentrations of vitamin $B_{12}$, illustrating the importance of vitamin $B_{12}$ synthesis in driving bacterial Co uptake (Osman et al., 2021); however, this mechanism has not been observed within marine bacterial communities. Additionally, vitamin $B_{12}$ uptake by both prokaryotes and eukaryotes has been found to be common in Antarctic coastal communities (Taylor and Sullivan, 2008; Rao, 2020), and likely contributes to the depletion of ligand-bound dCo in both the surface and mesopelagic ocean.

The shallower Zn apparent nutricline could also be explained by the higher stability of Zn metal-ligand complexes compared to Co complexes within phytoplankton metabolisms, allowing higher uptake rates of dZn when available (Irving and Williams, 1948; Sunda and Huntsman, 1995). The vertical dimension of trace metal loss captured by a comparison of these apparent nutriclines could be conceptualized as a time-dependent process driven by the phytoplankton community's preference for each trace metal, with preferred nutrients like Zn exhibiting a shallower stoichiometric inflection point arising from the rapid depletion of the metal within the photic zone, and nutrients like dCo, which is often taken up by eukaryotes when dZn is scarce (Sunda and Huntsman, 1995; Kellogg et al., 2020), exhibiting a deeper stoichiometric inflection point below the photic zone. This analysis suggests that substitution at the interface of the uptake



mechanism for trace metal transporters at least partially controlled the stoichiometry of Zn/Cd/Co
distributions and uptake in the upper ocean.
4.4 Zn/Cd/Co uptake using a shared trace metal membrane transport system
This study synthesized dissolved concentration and uptake datasets for Co, Zn and Cd
(Table 5), three trace metal nutrients whose use by phytoplankton is collectively integral to surface
ocean productivity and the biogeochemical cycling of Fe, vitamin $B_{12}$ and carbon in the Southern
Ocean. This combined dataset is ideal for interrogating questions of environmental competitive
inhibition of Zn, Cd and Co transport in low-dZn environments. The observation of order of
magnitude trends in trace metal uptake rates over depth profiles ($\rho$Zn > $\rho$Cd > $\rho$Co) was novel,
and paralleled the order of magnitude trends of trace metal concentrations in seawater ([Zn] > [Cd]
> [Co]; Fig 5). This environmental observation reflected the findings of numerous culture
experiments that quantify the uptake of trace metals as a function of the concentration of available
labile metals and the affinity of the metal for a cell transporter's binding ligand (Irving and
Williams, 1948; Sunda and Huntsman, 1992, 1995, 2000; Kellogg et al., 2020).
**Table 5.** Dissolved stoichiometric ratios and uptake stoichiometric ratios of five station profiles
for Co, Cd and Zn. The dCo : dCd : dZn : $dPO_4^{3-}$ ratio is the dissolved stoichiometry of metals
present in the water column normalized to $dPO_4^{3-}$, and the $\rho$Co : $\rho$Cd : $\rho$Zn ratio is the uptake
stoichiometry of microbial communities normalized to $\rho$Co.

| Region | Station | Depth [m] | dCo : dCd : dZn : $dPO_4^{3-}$ | $\rho$Co : $\rho$Cd : $\rho$Zn |
|---|---|---|---|---|
| Amundsen Sea | 4 | 10 | 19 : 314 : 1,716 : 1,000,000 | 1 : 8 : 56 |
| | | 30 | 23 : 295 : 1,889 : 1,000,000 | 1 : 16 : 88 |
| | | 50 | 24 : 293 : 2,096 : 1,000,000 | 1 : 15 : 108 |
| | 11 | 10 | 13 : 204 : 1,018 : 1,000,000 | 1 : 10 : 77 |
| | | 20 | 15 : 212 : 970 : 1,000,000 | 1 : 13 : 89 |
| | | 30 | 17 : 231 : 1,290 : 1,000,000 | 1 : 29 : 294 |
| | | 50 | 19 : 280 : 1,835 : 1,000,000 | 1 : 0*: 229 |
| | | 75 | 21 : 301 : 2,009 : 1,000,000 | 1 : 0* : 532 |
| | | 100 | 23 : 358 : 2,727 : 1,000,000 | 1 : 11 : 708 |
| | | 150 | 17 : 313 : 1,974 : 1,000,000 | 1 : 0* : 797 |
| | | 200 | 17 : 325 : 2,137 : 1,000,000 | 1 : 3 : 885 |
| Ross Sea | 20 | 30 | 17 : 323 : 1,846 : 1,000,000 | 1 : 13 : 349 |
| | | 50 | 13 : 305 : 2,020 : 1,000,000 | 1 : 11 : 163 |
| | | 100 | 17 : 333 : 2,400 : 1,000,000 | 1 : 0* : 376 |
| | | 150 | 16 : 330 : 2,321 : 1,000,000 | 1 : 0* : 507 |
| | | 200 | 14 : 336 : 2,358 : 1,000,000 | 1 : 0* : 913 |
| Terra Nova Bay | 22 | 10 | 15 : 136 : 617 : 1,000,000 | 1 : 12 : 81 |
| | | 25 | 10 : 158 : 546 : 1,000,000 | 1 : 11 : 75 |
| | | 40 | 11 : 254 : 1,127 : 1,000,000 | 1 : 29 : 166 |
| | | 75 | 13 : 301 : 1,965 : 1,000,000 | 1 : 7 : 228 |
| | | 100 | 11 : 304 : 1,938 : 1,000,000 | 1 : 56 : 705 |
| | | 150 | 16 : 310 : 2,212 : 1,000,000 | 1 : 122 : 584 |
| | 57 | 50 | 13 : 179 : 894 : 1,000,000 | 1 : 17 : 37 |
| | | 75 | 13 : 297 : 1,644 : 1,000,000 | 1 : 3 : 18 |
| | | 100 | 15 : 320 : 1,798 : 1,000,000 | 1 : 8 : 40 |

*Denotes depths at which $\rho$Cd was under the methodological detection limit.



Evidence for elevated Co uptake in the low-dZn environments of the surface ocean were
supported by the trace metal uptake rates. When $\rho$Zn and $\rho$Cd was normalized to $\rho$Co ($\rho$TM : $\rho$Co;
Fig. 5), deviations from these order-of-magnitude trends were observed; in particular, at Stations
4 and 11 in the Amundsen Sea and Station 22 in Terra Nova Bay, $\rho$Zn and $\rho$Cd stoichiometry
relative to $\rho$Co tended to decrease towards the surface in the upper 50 m, while the opposite trend
appeared to occur at Station 57 in the late summer. The surface-most trends of stations 20 and 57
were undetermined due to a lack of a 10 m $\rho$Co value. This increasing surface Co uptake
stoichiometry relative to Zn and Cd at Stations 4, 11 and 22 – stations that also displayed
significant phytoplankton blooms – suggests that Co uptake increased in low-Zn environments,
while later in the summer at Station 57, $\rho$Co lessened relative to $\rho$Zn, possibly due to the deepening
of the mixed layer in February, bringing additional dZn to the upper ocean via vertical mixing
(Fig. 4). The increase in the observed $\rho$Co rate was likely due to the upregulation of the shared Zn
and Co uptake transporter system.

From laboratory culture experiments aimed at examining the microbial uptake of Zn and
other trace metals, it is apparent that many diatoms and coccolithophores contain two distinct Zn
uptake systems: a low-affinity system that operates at higher concentrations of dZn and a high-
affinity system that functions at lower concentrations of dZn (Sunda and Huntsman, 1992; John et
al., 2007). Both transport mechanisms are relatively unspecific as to the divalent metals transported
into the cell; the low-affinity system is known to transport Zn, Cd and Mn, while the high-affinity
system transports Zn, Cd and Co. (Sunda and Huntsman, 1995, 1996); thus, Co uptake is often
inhibited at high dZn concentrations when the low-affinity system is active (Sunda and Huntsman,
1995; Sunda 2012). In culture, diatoms have been observed to switch from the low-affinity to the
high-affinity transport system between $10^{-10.5}$ and $10^{-9.5}$ M dZn$^{2+}$ (Sunda and Huntsman, 1992;
John et al., 2007), a relevant range for the lowest values of total dZn observed in the surface ocean
on CICLOPS (dZn minimum = $1 \times 10^{-10}$ M at Station 46, 10 m depth), and the dZn$^{2+}$ pool would
have been even smaller due to organic complexation.

To investigate the influence of transporter competitive inhibition on trace metal uptake via
the high-affinity uptake system, we can estimate the predicted $\rho$Co, $\rho$Cd and $\rho$Zn values given the
observed trace metal concentrations with an equation adapted from Michaelis-Menten enzyme-
substrate kinetics (Sunda and Huntsman, 1996, 2000):
$$\text{Predicted } \rho M = \frac{V_{max}[M^{2+}]K_M}{[Co^{2+}]K_{Co} + [Cd^{2+}]K_{Cd} + [Zn^{2+}]K_{Zn}}$$

where $M$ is the trace metal (Co, Cd, Zn) whose uptake is being calculated, $V_{max}$ is the saturation
uptake rate of the transporter system, and $K_{Co}$, $K_{Cd}$ and $K_{Zn}$ are steady state affinity constants for
the metal-ligand complex associated with the membrane transporter. For this system, we assumed
$K_{Zn} = K_{Cd} = K_{Co} = 10^{9.6}$, where $10^{9.6}$ is the value of $K_{Zn}$ for the high-affinity uptake system
determined by Sunda et al. (1992), and that 99% of the dCo, dCd and dZn inventory was bound to
strong organic ligands, leaving 1% of the total metal concentration labile. Note that the assumption
that $K$ and the percent labile multipliers are equal for all metals results in their value being nullified
by their presence in both the numerator and denominator of the predicted uptake equation, and so
their assumed values have no numerical impact on the predicted uptake values. It was also assumed
that $V_{max}$ values for each trace metal were equal, which is likely a reasonable assumption for metals
that share an uptake system, although $V_{max}$ is known to vary with trace metal concentration, a
function that we have assumed here to be negligible (Sunda and Huntsman, 1985, 1996; Sunda,





1989). $V_{max}$ is in units of μmol (mol C)$^{-1}$ d$^{-1}$, and the predicted trace metal uptake rates were
converted to units of M d$^{-1}$ using a C : Chl-a ratio of 130 w/w, derived from the Ross Sea
phytoplankton community (DiTullio and Smith, 1996).

When the predicted metal uptake rates were calculated using a $V_{max}$ value of 262 μmol (mol
C)$^{-1}$ d$^{-1}$ from previous Zn culturing experiments (Sunda and Huntsman, 1992), the resulting values
recreated the trend of the observed trace metal uptake profiles, with higher uptake rates in the
surface ocean and lower rates below the photic zone, but the predicted values were over an order
of magnitude greater than the measured uptake rates (Fig. A1). This offset may be due to several
factors: (1) the assumed C : Chl-a ratio to scale predicted uptake with observed biomass may be
high, (2) the $V_{max}$ value calculated from laboratory experiments may be high, or (3) the
assumptions that the speciation of the dissolved trace metals are 99% strongly-bound at all depths,
for all metals is incorrect. The final explanation may play a role in the offset between the predicted
and observed uptake rates, and illustrates the complexities of translating lab-based culture work to
environmental measurements and in-situ analyses. The $V_{max}$ value is also relatively unconstrained,
and it is reasonable to assume it may be lower in the Ross Sea than observed in culture if the
phytoplankton exhibit suppressed metal quotas to survive in a metal-deplete environment. With
this in mind, the $V_{max}$ value was tuned to 4 μmol (mol C)$^{-1}$ d$^{-1}$ to fit the observed uptake rates,
which is lower than any Co, Cd or Zn $V_{max}$ reported in the literature from culture studies (Fig. 10).
Using the tuned $V_{max}$ value, the high-affinity uptake system equation properly predicts the order
of magnitude trends inherent in the observed Co/Cd/Zn uptake rates. This analysis demonstrates
the measured uptake rates from the Ross Sea were likely driven by the concentration ratios of
available metals throughout the water column, following a high-affinity transporter model of Co,
Cd and Zn uptake.

The maximum diffusive limit, a calculation of the phytoplankton community's maximum
diffusion rate for the uptake of trace metal nutrients through their cell membranes, was also
estimated and compared to the observed and predicted uptake rate profiles. The physical limits of
uptake via diffusion was determined as a function of the surface area of phytoplankton membranes
(Sunda and Huntsman, 1992):

$$\text{Maximum diffusive limit} = 4\pi r D [M^{2+}]$$

where $r$ is the equivalent spherical radius of a phytoplankton cell, assumed to be 3 μm, a reasonable
value for diatom species, and $D$ is a diffusion rate constant of 2 x 10$^{-6}$ cm$^2$ s$^{-1}$, calculated for Zn$^{2+}$
at 20⁰C (Sunda and Huntsman, 1992). The diffusive limit was converted to units of M d$^{-1}$ using a
C : cell volume ratio of 12.5 mol C L$^{-1}$, which is the average of two diatom ratios reported in Sunda
and Huntsman (1995) (11 and 14 mol C L$^{-1}$), and the same C : Chl-a ratio of 130 w/w used for the
predicted uptake rate estimate above (DiTullio and Smith, 1996). The resulting diffusive limit
profiles are highly dependent on the assumed speciation of each trace metal; when the dCo, dCd
and dZn inventories were assumed to be 99% bound (Fig. 10), the maximum diffusive limit was
slightly greater than the predicted and observed uptake rates, but when the inventories were
assumed to be 100% labile (Fig. A1), the diffusive limit greatly exceeded the uptake rates by
several orders of magnitude. Since the metal inventories almost certainly vary in their speciation
of dZn and dCd over depth, as was observed in the dCo inventory, an accurate maximum diffusive
limit would exist between the two extremes of 0% bound and 99% bound, and might be expected
to be greater at deeper depths, where a higher fraction of the dissolved metal inventory is labile.
For additional analysis of the predicted metal uptake ratios and the maximum diffusive limit, see
Appendix H.



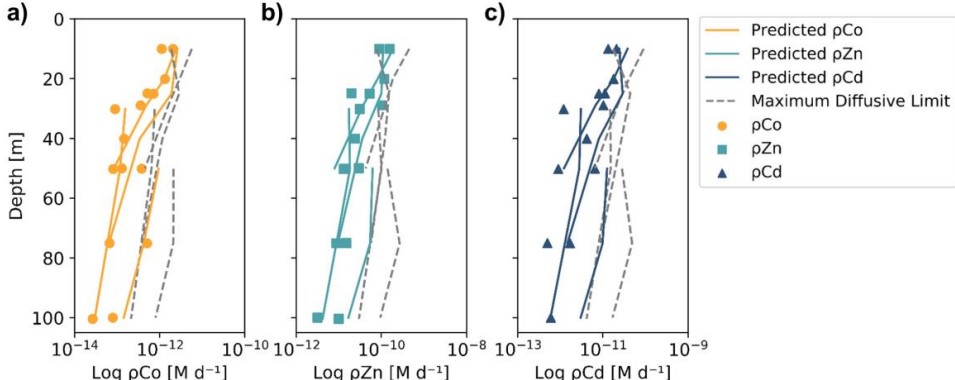

**Figure 10.** Observed (markers) and predicted (solid lines) trace metal uptake rate ($\rho$) profiles for
Co **(a)** Zn **(b)** and Cd **(c)** from Stations 11, 20, 22 and 57. The maximum diffusive limit profiles
(dashed lines) are shown as an estimate of the physical limits of metal diffusion through uptake
transporters. The predicted uptake rates were tuned to best fit the observed uptake rate trends by
using a $V_{max}$ value of 4 μmol (mol C$^{-1}$) d$^{-1}$, and the maximum diffusion limit estimation assumed
a speciation of 0.01% labile metals.

4.5 Vitamin B$_{12}$ and Zn stress, and their implications for increasing biological dCo demand

The near-absence of labile dCo and low concentration of ligand-bound dCo in coastal
Antarctic seas may indicate a larger shift in the region towards vitamin B$_{12}$ limitation. Vitamin B$_{12}$
has been shown to be co-limiting with Fe in the Ross Sea and elsewhere (Sañudo-Wilhelmy et al.,
2006; Bertrand et al., 2007), and increased vitamin B$_{12}$ uptake by both bacterioplankton and
eukaryotic phytoplankton has been observed in incubation experiments following the alleviation
of surface ocean Fe limitation (Bertrand et al., 2011). Two primary sources of Fe to the Antarctic
seas are a flux of lithogenic Fe from melting ice shelves along the continent and sediment
resuspension along the seafloor, both of which have been observed to be meaningful Fe sources to
the Amundsen Sea (Planquette et al., 2013; St-Laurent et al., 2017). The source of particulate Fe
from glacial meltwater to coastal Antarctic seas has been increasing over the past several decades
and is expected to continue to increase as Antarctic ice shelves and glaciers melt and retreat due
to global climate change (Monien et al., 2017). The source of particulate Co from glacial meltwater
would also be expected to increase since Co, like Fe, has been observed to be transported from the
Antarctic continent via ice melt (Westerlund and Öhman, 1991), and it is unclear what role this
presumably increasing source of Co to the surface ocean plays in the reduced inventories of dCo
in the surface ocean.

Although it is difficult to definitively conclude that the low dCo inventory observed on
CICLOPS is representative of a decadal trend towards vitamin B$_{12}$ limitation and not simply
variation in micronutrient availability and community structure, the inventory and stoichiometric
uptake trends documented in this study are compelling evidence for a changing biogeochemical
Co cycle in the coastal Southern Ocean. Paired with the recent discovery of Zn/Fe co-limitation in
Terra Nova Bay (Kellogg et al., [Submitted]), these results suggest a complex landscape of
micronutrient scarcity and limitation in coastal Antarctic seas where plankton community
structures and Fe additions from melting ice sheets can generate patches of vitamin B$_{12}$ and Zn
limitation within a broadly Fe-scarce HNLC region.



The bacterial community is essential to the development and alleviation of vitamin $B_{12}$ limitation within a eukaryotic phytoplankton bloom since only prokaryotes possess the metabolic pathway to synthesize the vitamin (Warren et al., 2002; Croft et al., 2005). In the Southern Ocean, near-zero counts of photosynthetic bacteria indicate that the heterotrophic bacterial communities are primarily responsible for vitamin $B_{12}$ production in the region (Bertrand et al., 2011). Vitamin $B_{12}$ can become limiting when the bacterial community is low in abundance and/or growth limited by a different nutrient such as dissolved organic matter (DOM). In the Ross Sea, bacterioplankton have been found to be growth limited by an inadequate supply of DOM (Church et al., 2000; Bertrand et al., 2011), and there can be up to a one-month lag between the onset of the spring phytoplankton bloom and an associated bacterial bloom stimulated by phytoplankton DOM production (Ducklow et al., 2001). This offset suggests that vitamin $B_{12}$ limitation among eukaryotes is most probable earlier in the season within the spring bloom. Additionally, low abundances of mesozooplankton and microzooplankton grazing rates in the Ross Sea create phytoplankton blooms with low grazing pressure (Caron et al., 2000; Ducklow et al., 2001), which may allow low DOM conditions to persist later into a bloom and exacerbate vitamin $B_{12}$ stress among eukaryotes.

A shift towards vitamin $B_{12}$ limitation would likely favor phytoplankton with flexible metabolisms that are able to reduce their demand for Co and vitamin $B_{12}$ when necessary. Organisms that can express the vitamin $B_{12}$-independent *metE* gene may out-compete those expressing the vitamin $B_{12}$-dependent *metH* gene (Rao et al., [In review]; Rodionov et al., 2003; Bertrand et al., 2013; Helliwell 2017). *P. antarctica*, for example, may be well suited to periods of vitamin $B_{12}$ limitation due to the symbiotic bacterial microbiomes that form within its colonies and produce B vitamins that allow the colonies to grow when B vitamins are otherwise unavailable (Brisbin et al., 2022). *P. antacrica* has also been found to express a novel *metE*-fusion gene when vitamin $B_{12}$ limited and *metH* gene while vitamin-replete, suggesting a highly flexible vitamin $B_{12}$ metabolism (Rao et al., [In review]).

There is compelling evidence for high rates of biological Co uptake in the Ross Sea during the 2017/2018 summer compared to the 2005/2006 summer driven by the uptake of dCo from vitamin $B_{12}$ and Zn scarcity. Together, these two stressors increase the rate of Co uptake as well as the Co : C stoichiometry of phytoplankton biomass. The stoichiometry of Co uptake has been observed to be highly plastic in this study and others, responding to the availability of other micronutrients and the requirements of the microbial community (Sunda and Huntsman, 1995; Saito et al., 2017). An increase in $\rho$Co could then result in a decrease of the Co inventory in coastal Antarctic seas, following the mechanism detailed below.

Biological uptake alone would not permanently remove Co from the water column; uptake only shifts Co from the dissolved phase to the particulate phase, where POM remineralization restores Co back to the dissolved phase. The net removal pathways of Co include (1) burial as POM, (2) particle scavenging and (3) depletion of dCo into Circumpolar Deep Water (CDW) and Antarctic Bottom Water (ABW). We have already noted that Co scavenging to Mn-oxides is particularly low in the Southern Ocean (Oldham et al., 2021). The advection of dCo into CDW may not be at a steady state throughout the year since cycles of ice melt and formation affect the mixing of CDW and formation of dense Antarctic Bottom Water (ABW), and so may represent a removal pathway for dCo on an annual cycle. However, an increase in the burial flux of Co in POM is the most likely pathway for sustained loss of the Co inventory. When the $\rho$Co rate increases, the stoichiometry of Co incorporation into biomass relative to P would also increase.





Over the years, a strengthened demand for Co via vitamin $B_{12}$ and Zn stress could result in a steady
loss of Co if the Co : C and Co : $PO_4^{3-}$ stoichiometry of POM increases but the remineralization of
POM is unchanged, increasing the flux of particulate Co into the deep ocean and sediments. In the
winter, sea ice covers the Antarctic seas and the water column mixes, a process that would
propagate the low dCo concentrations from the photic zone into the deep ocean and result in a
steady loss of the dCo inventory throughout the water column.
Additionally, warming surface ocean temperatures likely play a role in phytoplankton
productivity and nutrient uptake. Increasing both dFe availability and temperature have been
shown to significantly increase phytoplankton growth and phytoplankton abundance in the Ross
Sea, and impact community structure (Rose et al., 2009; Spackeen et al., 2018; Zhu et al., 2016).
From a kinetic perspective, higher surface temperatures would be expected to increase the uptake
rates of nutrients, including micronutrients like Fe, Co and Zn, by increasing the value of $K_M$.
However, the effects of temperature on productivity and community composition are more
complex since increasing ocean temperatures would also decrease the solubility of $CO_2$, change
the seasonality of ice cover and thus sunlight availability, and affect water column turnover and
mixing regimes (Rose et al., 2009). The effects of warming temperatures on the intricate landscape
of nutrient availability and limitation regimes described here is an open question in this study.
4.6 A two-box model that describes a mechanism for deep dCo inventory loss
To test the proposed mechanism that higher Co uptake rates and winter mixing can lead to
a deep inventory loss of ~10 pM Co over 12 years, a time step two-box model of a 1 $m^2$ water
column was created in Microsoft Excel to simulate the Ross Sea dCo cycle. A schematic of the
modeled dCo cycle is presented in Fig. 11, flux equations to describe the biogeochemical cycling
of Co are presented in Appendix I, and the parameters used to simulate dCo loss over 12 years and
a hypothetical steady state condition are given in Table 6.
The change in dCo concentration over time (d[dCo]/dt) for a surface ocean (0–100 m) and
deep ocean (100–500 m) was calculated as the sum of the dCo source fluxes minus the sum of the
sink fluxes:
$$\left(\frac{d[dCo]}{dt}\right)_{Surface} = \frac{F_{Over} + F_{Remin} - F_{Up}}{V_{Surface}}$$

$$\left(\frac{d[dCo]}{dt}\right)_{Deep} = \frac{F_{Remin} + F_{Neph} - F_{Up} - F_{Over}}{V_{Deep}}$$

where $F_{Over}$ is the overturning flux between the two boxes, $F_{Remin}$ is the remineralization flux, $F_{Up}$
is the biological uptake flux, and $F_{Neph}$ is the flux of dCo from the nepheloid layer into the deep
ocean (Table B1). $F_{Up}$ was calculated using the measured ρCo uptake rates observed on the
CORSACS and CICLOPS expeditions, and $F_{Remin}$ was calculated using an assumed surface and
deep remineralization factor (RF) of 0.9, indicating that 90% of the POM generated in the surface
ocean is remineralized back to its inorganic dissolved components. In the Southern Ocean, the
fluxes of scavenging and aerosol deposition would be relatively negligible, so these fluxes have
been omitted from the model. The magnitude of $F_{Neph}$ in the Ross Sea remains unconstrained, and
in this model, the deep nepheloid dCo source was used as an adjustable parameter to tune the
magnitude of deep dCo loss to be 10 pM over 12 years, which represents the approximate observed
differences between the CORSACS and CICLOPS expeditions detailed in Sect. 4.2. A $F_{Neph}$ was



calculated to be 3550 pmol $m^{-2}$ $d^{-1}$ to the deep ocean, but this should not be considered a
meaningful calculation of the observed nepheloid layer flux.

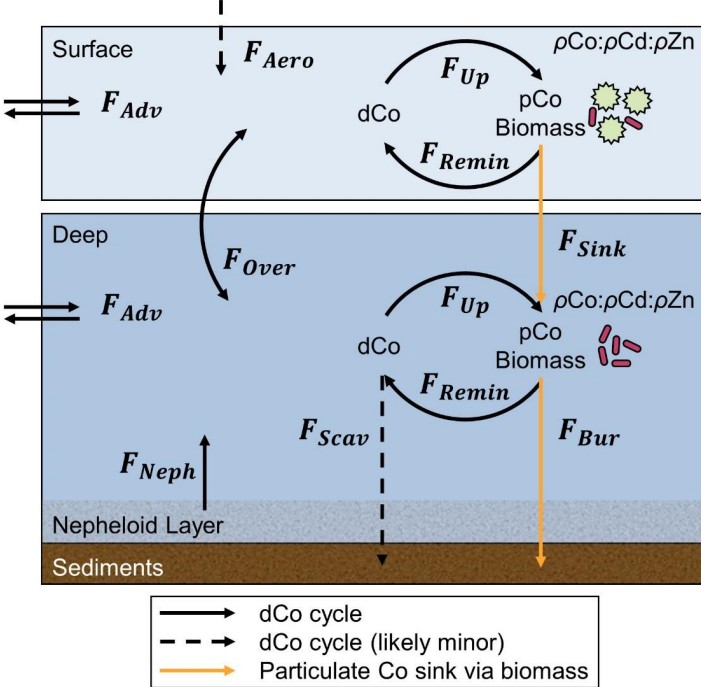

**Figure 11.** A schematic of the dCo cycle (black) and select processes of the particulate Co (pCo)
cycle (orange) presented as a simplified two-box model. Net fluxes of the dCo cycle include
sources from aerosol deposition ($F_{Aero}$), bottom sediments and the nepheloid layer ($F_{Neph}$), and
scavenging to Mn-oxides particles ($F_{Scav}$) which likely represents a minor flux in the coastal
Antarctic seas. Internal cycling fluxes include horizontal advection ($F_{Adv}$), water column
overturning or mixing ($F_{Over}$), biological uptake ($F_{Up}$) and remineralization of pCo ($F_{Remin}$). Fluxes
of pCo shown here include sinking biomass from the surface into the deep ocean ($F_{Sink}$) and pCo
burial into sediments along the seafloor ($F_{Bur}$). The biological uptake of dCo is influenced by the
relative stoichiometric uptake of Co, Zn and Cd ($\rho$Co : $\rho$Cd : $\rho$Zn) among the microbial
community. Differential equations that describe and quantify these fluxes are presented in
Appendix I.

In the Ross Sea, the deep winter mixed layer can extend 600 m to the seafloor and turn
over the whole water column in some locations (Smith and Jones, 2015), mixing the surface and
deep ocean under the winter sea ice and resulting in near-vertical profiles of dCo in the early spring
(Noble et al., 2013). Here, the winter mixing process was modeled by combining the surface and
deep ocean boxes into one homogenized box during the winter season (151 days, ~5 months). The
dCo concentrations of the winter box were calculated using a volume-weighted average (see
Appendix I).
**Table 6.** Parameters of the Co cycle two-box model, run as both a steady state model with lower
Co uptake rates ($\rho$Co) and as a mechanism for deep dCo inventory loss driven by higher $\rho$Co
values. The calculated burial flux of particulate Co within each model variation is also given, but



note that the burial flux values should be interpreted as a comparison of the Co sink via the
biological pump when $\rho$Co is varied, and not as observed or meaningful Co flux magnitudes.

| Model Parameters | Value | Units |
|---|---|---|
| Bloom season length | 214 | days |
| Surface box height | 100 | m |
| Deep box height | 500 | m |
| Remineralization Factor (RF) | 0.9 | |
| Deep Nephloid Flux | 3550 | pmol Co m$^{-2}$ d$^{-1}$ |
| Overturning Water Flux | 0 | m$^3$ d$^{-1}$ |
| **Steady State Parameters** | | |
| Surface $\rho$Co | 0.27 | pmol Co L$^{-1}$ d$^{-1}$ |
| Deep $\rho$Co | 0.66 | pmol Co L$^{-1}$ d$^{-1}$ |
| Burial Flux | 3550 | pmol Co m$^{-2}$ d$^{-1}$ |
| **dCo Loss Parameters** | | |
| Surface $\rho$Co | 0.87 | pmol Co L$^{-1}$ d$^{-1}$ |
| Deep $\rho$Co | 0.1 | pmol Co L$^{-1}$ d$^{-1}$ |
| Burial Flux | 5870 | pmol Co m$^{-2}$ d$^{-1}$ |


This model provides a plausible mechanism by which increases in $\rho$Co such as those
observed along the CICLOPS expedition might increase the burial flux of particulate Co, resulting
in a net loss to the deep dCo inventory. The uptake rate of Co both within and below the photic
zone, as well as the fraction of POM that is remineralized, dictated the flux of particulate Co into
the sediments via burial. The initial dCo concentration was set at 56 pM, which approximates the
mean deep dCo concentrations observed on both CORSACS-1 and CORSACS-2. When the model
was run for 12 years, the time period between the first CORSACS expedition and the CICLOPS
expedition, it generated a sawtooth pattern; the surface and deep boxes diverged over the course
of the summer bloom season as biological uptake removed dCo from the surface box and
remineralization replenished dCo in the deep box (Fig. 12). Winter mixing then unified and reset
the water column, replenishing the surface dCo inventory. The model was run at a steady state
using the average surface $\rho$Co rate observed on CORSACS-1 (0.27 pmol L$^{-1}$ d$^{-1}$; Table 6) (Saito
et al., 2010) and deep $\rho$Co values that were tuned to allow no change in the deep dCo inventory
every winter. When the model was run using representative surface and deep $\rho$Co values observed
on the CICLOPS expedition (0.87 and 0.1 pmol Co L$^{-1}$ d$^{-1}$, respectively), the surface depletion of
dCo was more pronounced by the end of the bloom season compared to the steady state model,
and winter mixing resulted in a steady annual decrease of the deep dCo inventory. The mechanism
of dCo loss was driven by increasing $\rho$Co, particularly in the surface ocean, and the propagation
of dCo loss into the deep ocean via vertical mixing. The resulting burial flux when the model
exhibited a deep dCo loss mechanism was higher than when the model was run at a steady state
(Table 6), demonstrating how higher Co uptake rates among plankton paired with a deep winter
mixed layer can result in a diminishing dCo inventory on a decadal timescale.



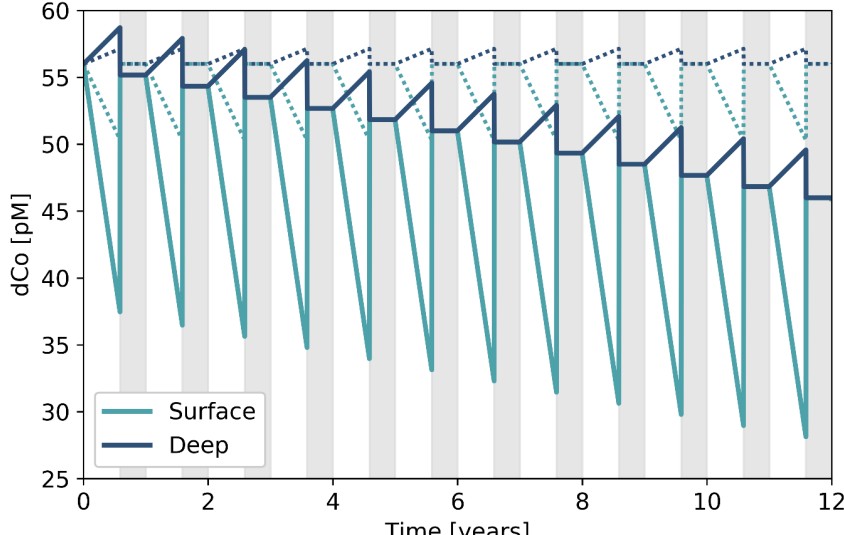

**Figure 12.** Results of the two-box model illustrating a potential mechanism for the loss of the dCo inventory over time. Gray boxes represent the winter season when the surface and deep boxes mix. The dotted lines represent a system at a steady state, where the dCo inventory stays consistent annually. The solid lines represent a system exhibiting dCo loss, where increased Co uptake rates in both the surface and deep ocean result in an annually decreasing dCo inventory. The initial deep dCo concentration was 56 pM, which approximates the mean deep dCo concentrations observed on CORSACS-1 and CORSACS-2. Over 12 years, the dCo loss model depicts the loss of 0.83 pM year$^{-1}$ to end at a deep dCo inventory of 46 pM, the mean deep dCo concentration observed on CICLOPS.

The purpose of this model was to illustrate a possible mechanism for a dCo inventory loss over the 12-year period between the CORSACS and CICLOPS expeditions using reasonable estimates of Co uptake and other Co cycle fluxes to achieve the observed 10 pM deep inventory loss. This box model successfully shows the directionality of the changes to the deep ocean dCo inventory and deep burial flux when the $\rho$Co values increase, but the magnitude of the estimated Co burial or the nepheloid Co source should not be considered meaningful flux values. The model represented a greatly simplified version of the carbon pump in the Southern Ocean, and it is likely that at least some of the unquantified Co cycle fluxes were not negligible, including horizontal advection, overturning water during the summer season, Co scavenging, and a surface aerosol source. Additionally, it is a simplifying assumption that $\rho$Co values would be consistent throughout a surface or deep depth region, as well as consistent over an entire summer season. Despite its simplicity, the box model presented a concise and reasonable mechanism for this study's observation of a shrinking dCo inventory in the Ross Sea.

**5 Conclusion**

The Ross Sea, Amundsen Sea and Terra Nova Bay displayed lower dCo and labile dCo inventories during the 2017/2018 austral summer relative to prior observations in the region, which is consistent with observations of higher rates of Co use and uptake by phytoplankton and



heterotrophic bacteria. The near-100% complexation of the dCo inventory reveals that the dCo loss is primarily due to the uptake of labile dCo, the most bioavailable form of dCo to marine microbes. The decrease in dCo throughout the water column compared to prior observations is indicative of a multi-year mechanism, whereby the removal of dCo from the surface mixed layer via uptake over the summer has been propagated into the deep ocean via winter mixing, resulting in a decrease in dCo concentration throughout the water column. This change may be due to the alleviation of Fe limitation through inputs from increased glacial melting and subsequent development of intermittent vitamin $B_{12}$ and/or Zn limitation, both of which would be expected to increase the demand for Co among plankton communities.

In coastal Antarctica and other regions impacted by global climate change, Co is a noteworthy trace metal nutrient to investigate because its small inventory and flexible phytoplankton stoichiometry make its biogeochemical cycle particularly vulnerable to perturbation. In the Arctic Ocean, for example, the dCo and labile dCo inventories have increased as melting ice and permafrost have increased the flux of Co-enriched riverine waters and sediments to the upper ocean (Bundy et al., 2020). Like many other trace nutrients, the Co cycle is integrally connected to that of other elements like Zn, Cd, Fe and carbon, and observations of perturbed Co inventories and changing nutrient limitation regimes would affect their biogeochemical cycles as well. In highly productive coastal Antarctic seas, shifts in micronutrient inventories and growth limitation could have implications for the composition of regional phytoplankton blooms and the magnitude of the Southern Ocean carbon sink.

Since the late 1980s, it has been hypothesized that the primary productivity and net carbon sequestration flux of the Southern Ocean is controlled by the supply of Fe to surface waters (Martin 1990; Martin et al., 1990). This theory, called the "iron hypothesis", posits that the addition of bioavailable Fe to an Fe-limited surface ocean stimulates productivity and, in turn, increases the regional and possibly global carbon sequestration flux from the atmosphere into deep ocean sediments. When applied to potential carbon dioxide removal (CDR) geoengineering projects, the iron hypothesis provides a theoretical framework for ocean iron fertilization (OIF), where significant quantities of Fe are introduced to the surface Southern Ocean to enhance the net sequestration of $CO_2$ and reduce global atmospheric $CO_2$ concentrations (Emerson, 2019). Over the past three decades, several mesoscale Fe fertilization experiments have shown that large phytoplankton blooms can be stimulated by the addition of Fe to the surface Southern Ocean, and that the impact on the $CO_2$ sink is variable, modest and often difficult to assess (Coale et al., 1996; Boyd et al., 2000; de Baar et al., 2005; Smetacek et al., 2012). This study provides additional insights into the potential of OIF, suggesting that the alleviation of Fe limitation might shift the region towards the limitation of another trace nutrient such as vitamin $B_{12}$, Zn, and potentially Co. The nutrient limitation regimes of the Southern Ocean are complex, heterogeneous and possibly shifting on decadal timescales, and these intricacies must be examined when considering future OIF projects.

**Appendix A. Estimating trace metal uptake and maximum rate of dissolution profiles from classic competitive inhibition equations.**

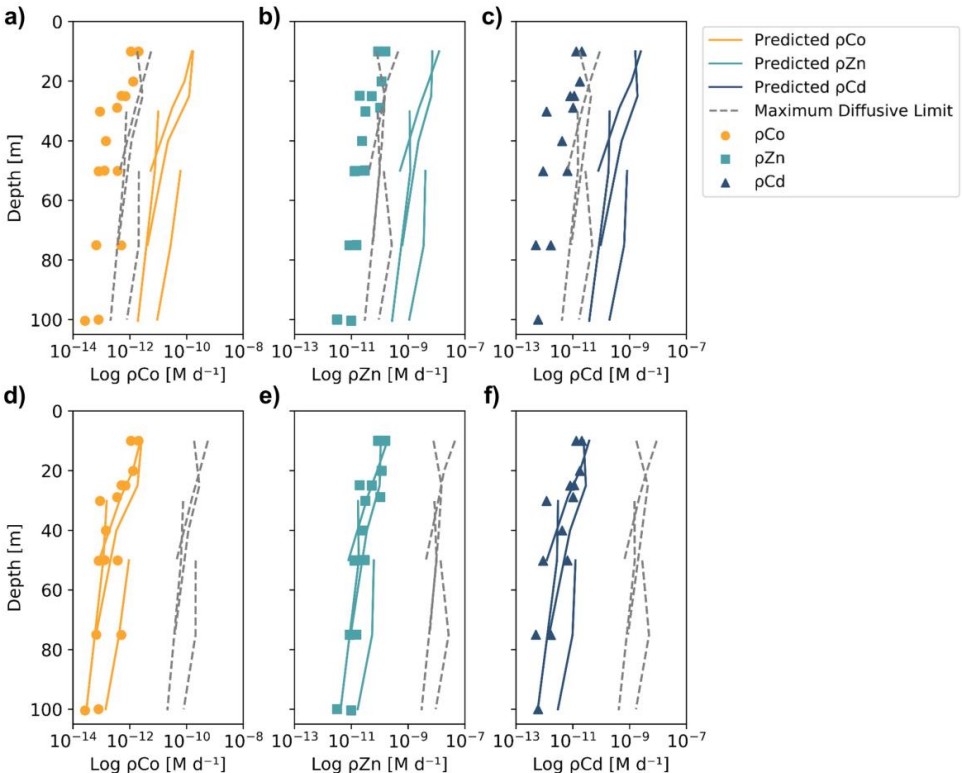

**Figure A1.** Observed (markers) and predicted (solid lines) trace metal uptake rate ($\rho$M) profiles and the estimated maximum diffusive limit profiles (dashed line) for Co **(a,d)** Zn **(b,e)** and Cd **(c,f)** from Stations 11, 20, 22 and 57, using different equation parameters than those used in Fig. 10. In panels a-c, the predicted uptake rates used a literature $V_{max}$ value of 262 µmol (mol C$^{-1}$) d$^{-1}$ determined from Zn$^{2+}$ uptake experiments in *Emiliania huxleyi* cultures (Sunda and Huntsman, 1992), resulting in predicted uptake rates that were orders of magnitude greater than the observed values. In panels d-e, the estimated maximum diffusive limit profiles assumed that 100% of the dCo, dZn and dCd inventories were labile and 0% were bound to strong organic ligands, resulting in diffusive limits that were also orders of magnitude greater than the observed values. This analysis helps to show how parameter assumptions can greatly influence the predicted uptake rates and illustrates the difficulty of assigning kinetic parameters to environmental analyses.

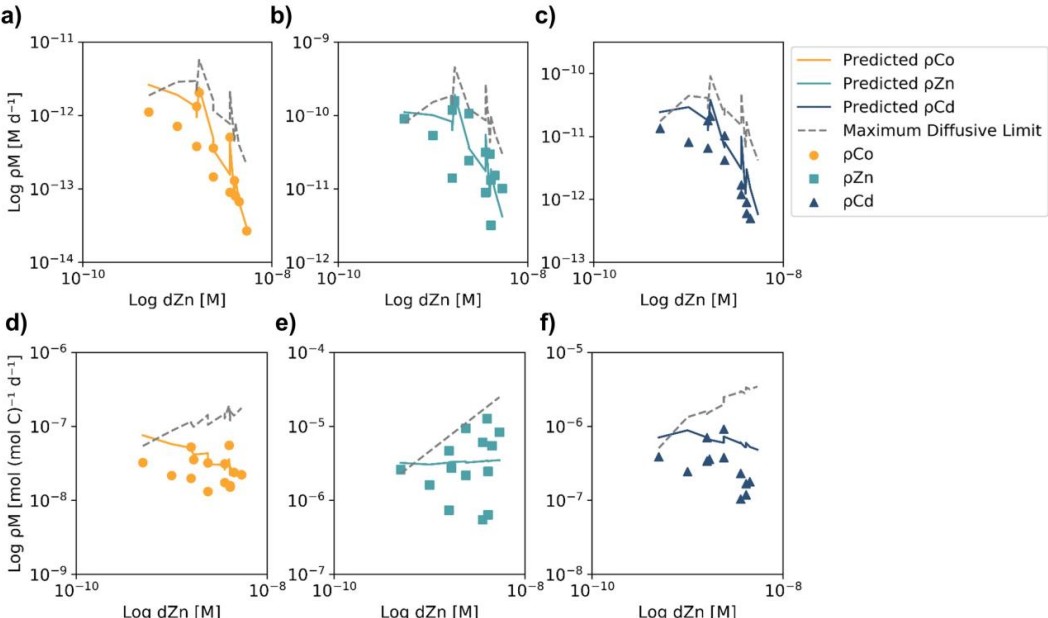

1070

**Figure A2.** Observed (markers) and predicted (solid lines) trace metal uptake rates ($\rho$M) and the estimated maximum diffusive limit profiles (dashed line) plotted against total dZn concentrations, assuming a $V_{max}$ of 4 μmol (mol C$^{-1}$) d$^{-1}$ and that 99% of the trace metal inventory was bound to strong organic ligands. Panels **a-c** show $\rho$M in units of M d$^{-1}$, which tended to decrease at high dZn concentrations. This is attributable to higher dZn concentrations below the photic zone, where much lower rates of micronutrient uptake occur. Panels **d-f** show $\rho$M when normalized to biomass using Chl-a concentrations and a C : chl-a ratio of 130 w/w (DiTullio and Smith, 1996). The normalized predicted $\rho$Zn values are relatively stable over the observed range of dZn concentrations, while the predicted $\rho$Co and $\rho$Cd values decrease slightly as dZn increases, suggesting that competitive inhibition of $\rho$Co and $\rho$Cd may have occurred at higher dZn concentrations due to the smaller inventories of dCo and dCd compared to dZn.

1082



**Appendix B. Description of a two-box model of the dCo cycle in coastal Antarctic seas, and a potential mechanism for deep dCo loss with changing microbial uptake stoichiometry.**

The two-box model described below was used to conceptualize the biogeochemical cycling of dCo in the surface and deep ocean. The model describes a 1 m$^2$ column of water with a total depth of 600 m and a depth threshold between the surface and deep box of 100 m. Within each box, the net change of dCo over time is equivalent to the sum of the source fluxes minus the sum of the sink fluxes:

$$\left(\frac{d[dCo]}{dt}\right)_{Surface} = \frac{\sum(F_i)_{Sources} - \sum(F_i)_{Sinks}}{V_{Surface}}$$

$$\left(\frac{d[dCo]}{dt}\right)_{Deep} = \frac{\sum(F_i)_{Sources} - \sum(F_i)_{Sinks}}{V_{Deep}}$$

where fluxes ($F_i$) are in units of mols dCo d$^{-1}$. A summary of the sources and sinks relevant to dCo in coastal Antarctic seas is shown below in Table B1. In the Southern Ocean, we would expect the fluxes of scavenging ($F_{Scav}$) and aerosol deposition ($F_{Aero}$) would be relatively negligible, and so these fluxes have been omitted from the model. Additionally, we can assume that horizontal advection is at a steady state, and thus the net advection flux is $\approx 0$ mols dCo d$^{-1}$. This gives us the net equations for both boxes:

$$\left(\frac{d[dCo]}{dt}\right)_{Surface} = \frac{F_{Over} + F_{Remin} - F_{Up}}{V_{Surface}}$$

$$\left(\frac{d[dCo]}{dt}\right)_{Deep} = \frac{F_{Remin} + F_{Neph} - F_{Up} - F_{Over}}{V_{Deep}}$$

**Table B1:** The source and sink fluxes of dCo in the surface and deep ocean boxes. Fluxes are theoretically in units of mols dCo d$^{-1}$.

| Surface Sources | | Surface Sinks | | Deep Sources | | Deep Sinks | |
| --- | --- | --- | --- | --- | --- | --- | --- |
| Remineralization | $F_{Remin}$ | Microbial Uptake | $F_{Up}$ | Remineralization | $F_{Remin}$ | Microbial Uptake | $F_{Up}$ |
| Overturning | $F_{Over}$ | Advection | $F_{Adv}$ | Nepheloid Layer | $F_{Neph}$ | Overturning | $F_{Over}$ |
| Aerosols | $F_{Aero}$ | | | Advection | $F_{Adv}$ | Scavenging | $F_{Scav}$ |
| Advection | $F_{Adv}$ | | | | | Advection | $F_{Adv}$ |

Uptake fluxes

The flux of dCo incorporation into microbial biomass via uptake by protein transporters can be described using the uptake rates ($\rho$Co) measured by $^{57}$Co incubation experiments, where units of $p$Co are in mols dCo L$^{-1}$ d$^{-1}$:

$$F_{Up,Surface} = (\rho Co_{Surface} * V_{Surface})$$

$$F_{Up,Deep} = (\rho Co_{Deep} * V_{Deep})$$





Remineralization fluxes

In this model, the remineralization flux of particulate Co in organic matter to dCo is quantified by a Remineralization Factor (RF), which can be applied to the amount of particulate matter present in each box. Typical RF values tend to be between 0.90 and 0.99 (Glover et al., 2011), meaning that between 90% and 99% of all microbial biomass produced tends to be remineralized before sinking out of its respective box. It is not clear that the RFs for the surface and deep box should be represented by the same value, and so we have defined both surface ($RF_{Surface}$) and deep ($RF_{deep}$) variables here. In the surface ocean, excess Co in un-remineralized biomass will sink into the deep box ($F_{Sink}$), where it is further able to be remineralized in the deep ocean. In the deep ocean, excess Co in un-remineralized biomass is assumed to flux into the sediments via burial ($F_{Bur}$), representing a key sink of dCo biomass out of the two-box system. The surface box remineralization flux is represented with a relatively simple equation:

$$F_{Remin,Surface} = RF_{Surface} * F_{Up,Surface}$$

$$F_{Remin,Surface} = RF_{Surface}(\rho Co_{Surface} * V_{Surface})$$

The deep ocean remineralization flux can then be calculated as the sum of the remineralization flux from excess biomass that sinks as particulate Co and biomass generated in the deep ocean:

$$F_{Remin,Deep} = RF_{Deep}(F_{Up,\,Surface} - F_{Remin,Surface}) + RF_{Deep}(F_{Up,\,Deep})$$

$$F_{Remin,Deep} = RF_{Deep}(\rho Co_{Surface} * V_{Surface} - RF_{Surface}(\rho Co_{Surface} * V_{Surface}))$$
$$+ RF_{Deep}(\rho Co_{Deep} * V_{Deep})$$

Overturning fluxes

An overturning dCo flux represents the flux of a volume of water from the deep ocean box into the shallow ocean box, and a corresponding flux of the same volume from the shallow ocean box into the deep ocean box for mass conservation. In a dynamic coastal upwelling system like the Ross and Amundsen Seas, the reality of this overturning flux is almost certainly much more complicated, as coastal upwelling processes overlap with meltwater processes and deep water mass formation processes. For the purposes of this two-box model, the flux of dCo via overturning can be estimated as a function of the overturning water flux ($F_{Water}$) and the dCo concentrations of each box:

$$F_{Over,Surface} = (F_{Water}[dCo]_{Deep} - F_{Water}[dCo]_{Surface})$$

$$F_{Over,Deep} = (F_{Water}[dCo]_{Surface} - F_{Water}[dCo]_{Deep})$$

In the model presented in Sect. 4.6, the $F_{Water}$ and both $F_{Over}$ fluxes are assumed to be negligible for the sake of modeling simplicity, but the introduction of a nonzero overturning flux would help to make the seasonal change in the dCo inventory in both the surface and deep oceans nonlinear, as it is currently the only flux in this model that is calculated using the time step's dCo concentrations.

Flux from the nepheloid layer

At several CICLOPS stations, a distinct nepheloid layer was detected as dCo concentration increased sharply at depths immediately above (~10 m) the ocean floor. The nepheloid layer tends



to contain high levels of particles moving horizontally along the seafloor, and is likely a significant source of dCo to the surrounding water column. The source of dCo from the nepheloid layer is somewhat unclear; it could be via dissolution of particles suspended within the nepheloid layer or from a porewater flux of dCo out of the sediments. In this model, the flux of deep dCo inputs into the deep ocean, assumed to be from the nepheloid layer, was derived using the Microsoft Excel solver tool, given the parameter that 10 pM of deep dCo was lost over 12 years. The deep source of dCo was calculated to be 3550 pmol dCo m$^{-2}$ d$^{-1}$. This value should be considered an adjustable parameter used to tune the model to our conceptual understanding of dCo inventory loss, and not a meaningful calculation of observed Co flux from the deep nepheloid layer, which has yet to be constrained.

The cobalt burial sink

The loss of cobalt from the deep ocean box into the sediments via burial can be quantified with the equation:

$$\left(\frac{d[Co]}{dt}\right)_{Bur} = F_{Sink} + F_{Up,Deep} - F_{Remin,Deep}$$

where $F_{Sink}$ is described by:

$$F_{Sink} = (\rho Co_{Surface} * V_{Surface}) - RF_{Surface}(\rho Co_{Surface} * V_{Surface})$$

This estimate of the loss of dCo due to burial assumes that all biogenic particulate Co that is not remineralized in the surface ocean sinks into the deep ocean, and all biogenic particulate Co that is not remineralized in the deep ocean is sequestered in sediments and "lost" to the model.

Modeling seasonality: the winter mixed layer

In the Ross and Amundsen Seas, sea ice covers the surface ocean for a larger portion of the year (~ 5 months). During this time, the water column mixes – a process that was modeled by combining the two-box model into one homogenized box after the 7-month bloom season to simulate the winter season. This process can be modeled by a volume-weighted average with the volume of each box.

$$[dCo]_{Winter} = \frac{(V_{Surface} * [dCo]_{Surface}) + (V_{Deep} * [dCo]_{Deep})}{(V_{Surface} + V_{Deep})}$$

**Data availability**

The CICLOPS dCo dataset has been submitted to the Biological and Chemical Oceanography Data Management Office (BCO-DMO) website (https://www.bco-dmo.org/project/774945) and is pending approval for publication. The dissolved metals (dZn, dCd) dataset (https://www.bco-dmo.org/dataset/877466), Zn and Cd uptake rate dataset (https://www.bco-dmo.org/dataset/877681), and macronutrient dataset (https://www.bco-dmo.org/dataset/874841) are publicly available on the BCO-DMO website.

**Author contribution**

RC collected and analyzed dCo samples and wrote the manuscript. RK collected and analyzed dZn and dCd samples and measured Zn and Cd uptake rates. DR measured Co uptake rates. GD



collected and analyzed phytoplankton pigment samples. All authors assisted in the collection and processing of dissolved seawater samples and incubation experiment samples, and all authors helped write the manuscript.

**Competing interests**

The authors declare that they have no conflict of interest.

**Acknowledgments**

The authors thank the captain, crew and science party of the RVIB *Nathaniel B. Palmer* for their support during the CICLOPS expedition. We also thank Joe Jennings (OSU) for dissolved macronutrient analysis, Véronique Oldham for sampling assistance, and Matthew Charette, Stephanie Dutkiewicz, and Alessandro Tagliabue for writing insights. This work was funded by grants from the National Science Foundation's Office of Polar Programs (OPP-1643684, OPP-1644073 and OPP-1643845).

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
