# Peer review of "Low Cobalt Inventories in the Amundsen and Ross Seas Driven by High Demand for Labile Cobalt Uptake Among Native Phytoplankton Communities"

_EGUsphere, 2023_

## Author Comment (AC1)

**Reviewer 1: Comments and Author Response**

The manuscript titled, "Low Cobalt Inventories in the Amundsen and Ross Seas Driven by High Demand for Labile Cobalt Uptake Among Native Phytoplankton Communities" written by Rebecca J. Chmiel and coauthors, describes how dissolved cobalt concentrations in the Ross Sea were much lower in the 2017-2018 season compared to two previous expeditions a decade earlier. The differences in these observations were explored by examining dissolved cobalt (dCo), zinc and cadmium uptake rates, as well as dCo vs. phosphate relationships to gain insights into processes acting on the dCo pool. Overall, I really enjoyed reading this manuscript and thought it was very thought-provoking and thorough.

Most of my comments below are very minor and driven primarily by interest. My only more general question for the authors was whether they had explored potential differences in the water masses sampled during the CICLOPS and CORSACS expeditions, which may impact the deep dCo concentrations. For example, if the expeditions both appeared to sample Circumpolar Deep Water (CDW) and Antarctic Bottom Water (AABW) equally well, then that would strengthen the argument that the dCo inventory differences are primarily driven by differences in Co uptake by the phytoplankton community rather than changes in the water masses over the Amundsen shelf or in the Ross Sea.

> [Authors' Response] A comparison of the water masses sampled during all 3 expeditions presented in this paper would be an asset to its overall conclusion. We have added a figure in the appendix (Fig. A2) to compare the hydrography and water mass classification of the samples used to show differences in the deep dCo inventory, and included a sentence in Sect. 4.3 that reads: "Since a plot of temperature vs. salinity shows largely overlapping hydrography among the three expeditions in the Ross Sea (Fig. A2), the observed difference in dCo inventories is unlikely to be due to differences in the distributions of the water masses sampled."

Perhaps related to this, I was also wondering if the authors have any evidence that the potential for Mn-oxidation might have changed over the 2007-2018 time period, perhaps due to a change in temperature? It seems like Mn-oxidation is low to non-existent in this region, but perhaps it would be something to think about for future work in this area and might have a significant impact on Co scavenging.

> [Authors' Response] While an examination of changing Mn-oxidation rates over time is certainly intriguing, it falls outside the scope of this study. An in-depth examination of Mn speciation and Mn oxidation on the CICLOPS expedition is presented in Oldham et al., 2021, where the authors do compare observed Mn concentration and speciation to previous studies, including the 2006 McMurdo Sound fieldwork discussed here.

In general, I thought this was an excellent paper and it will be an exciting contribution to the field. Below are some additional very minor more specific comments.

**Specific comments**

Figure 1: Can you perhaps note broadly on this figure where the CORSACS cruises were? I realize it might clutter the figure to have all of the stations, but maybe just an outline of the regions that those cruises sampled?

> [Authors' Response] I believe it does clutter Figure 1 to add the historical CORSACS and McMurdo Sound data. Instead, I have added Figure A1 to a new Appendix A, which maps the CORSACS-1, CORSACS-2, and McMurdo sound data onto the original maps for the CORSACS data to show how they overlap.

Figure 4 and 7: How were the regression outliers selected?

> [Authors' Response] In Figure 4, the one outlier in dZn concentration was selected by hand as a datapoint that did not fit the otherwise smooth dZn profiles analyzed, although the outlier was retained as shown in case this high signature is a true feature at this depth.

> In Figure 7, outliers were selected by hand when including them in the linear regression substantially decreased its $R^2$ value. Often, this was due to upper ocean samples at some stations sharing similar properties as the deep ocean samples (for example, the Ross Sea dZn vs. $dPO_4^{3-}$ subplot). We did our best to select depth thresholds that best described the data by region, but combining multiple stations in one regional graph will always be tricky since not all stations in a region share the same hydrography.

> To clarify how the outliers were selected, a sentence in the legend of Figure 7 has been modified to read: "Regression outliers were selected by hand when including them in the linear regression substantially decreased its $R^2$ value; outliers are marked with an 'x'."

Figure 6: I thought it was interesting that the CICLOPS expedition shows more of a scavenging signal for dCo compared to the CORSACS expedition. Any thoughts on why there might be those differences?

> [Authors' Response] Although dCo profiles for some stations on CICLOPS show a decreased dCo concentration in the mesopelagic, as we would expect with a scavenging signal (Fig. 2 Station 22, for example), the trend of lower dCo with respect to $dPO_4^{3-}$ is not visible in the dCo vs. $dPO_4^{3-}$ plot (Fig. 6b), indicating that this decrease may not be attributable to scavenging.

Figure 9: I really like this figure, the trends are very clear and it is really interesting.

> [Authors' Response] Thank you!

Line 731-736: Perhaps split this into multiple sentences.

> [Authors' Response] This change has been made. The passage now reads: "When $\rho$Zn and $\rho$Cd was normalized to $\rho$Co ($\rho$M : $\rho$Co; Fig. 5), deviations from these order-ofmagnitude trends were observed. In particular, at Stations 4 and 11 in the Amundsen Sea and Station 22 in Terra Nova Bay, $\rho$Zn and $\rho$Cd stoichiometry relative to $\rho$Co tended to decrease towards the surface in the upper 50 m, while the opposite trend appeared to occur at Station 57 in the late summer."

Figure 12: Is it possible to also plot an average of the dCo in the deep and surface box over time on top of the evolution of the pools? I thought it would be interesting to see if this dCo loss is a steady decrease or not, based on this model. It appears to be steady based on the trends in the deep and surface, but seeing the average plotted on top of this might be interesting.

[Authors' Response] An average dCo concentration for each austral summer year could be plotted over this figure, but I would argue it is not necessary. The model presented is linear and has been set to show a loss of 10 pM in the deep ocean over a 12-year period. I suspect adding an average rate of loss trendline would both clutter the figure and imply that the rate of loss was a calculated outcome of the model, instead of an input parameter.

---

## Author Comment (AC2)

**Reviewer 2 Comments and Author Response**

Rebecca J. Chmiel and coauthors present an original, high-quality investigation of why the dissolved cobalt inventory of specific coastal Antarctic seas has decreased between 2005/6 and 2017/18. They achieve this by comprehensively exploring cobalt speciation, stoichiometry, and uptake rates alongside that of zinc and cadmium, trace metals with inter-related ocean biogeochemical cycles. With climate induced changes in the cycling of trace metals related to that of carbon, this decadal-scale study is both timely and thoughtful. I thank the authors for bringing this thought-provoking work together and congratulate them on the quality of the paper. Overall, I believe this study aligns well with the aims/scope of BG and will be of clear interest to the marine biogeochemical community.

My specific comments are relatively minor but reflect aspects of the paper I thought could be improved/clarified.

**Specific Comments**

Whilst this is an intriguing investigation of the coastal Antarctic Co cycle, the important big picture insight I took from this paper was that climate driven changes in the Antarctic coastal Fe cycle may be responsible for dramatic regional shifts in micronutrient biogeochemistry and limitation patterns. This point could be emphasized more in the introduction, without the need for drastic changes to the text. At present there is one sentence at the end, broadly describing shifts in Zn/B12 limitation as a result of a warming climate, but with no mention of the influence of Fe supply.

> [Authors' Response] In this study, we have been careful to clarify that we only observed changes to the dCo cycle, including changes in vitamin B12 uptake, and dCo concentrations relative to other trace nutrients. While we believe that these changes may be linked to perturbations in the dFe cycle, this finding is speculative and has not been confirmed.
>
> However, we have modified the final sentence of the introduction to clarify our hypothesis that the observed perturbations may be due to a changing dFe cycle. The sentence now reads: "The results presented by this study reveal a substantial perturbation of the Co cycle, a shift towards vitamin $B_{12}$ and/or Zn limitation, and possible, but unconfirmed, perturbations to the dissolved iron (dFe) cycle in coastal Antarctic waters impacted by high rates of glacial ice melt and a warming climate."

Was dZn measured during CORSACS? If Zn limitation is in part driving the dCo inventory differential between CORSACS and CICLOPS, it would be interesting to read whether there has been a change in dZn concentration over this same time period?

> [Authors' Response] The discovery of Zn limitation in Terra Nova Bay and the change in dZn concentration over time are highly interesting phenomena discovered during the CICLOPS Expedition that deserves its own paper. This journal article is Kellogg et al.,

(In Review), as well as Kellogg et al., (2022) – chapter 6 of Dr. Riss Kellogg's thesis (https://dspace.mit.edu/handle/1721.1/144738).

Could you say something about the broad pattern of sampling during CICLOPS at the start of the methods section? Where samples taken sequentially from Amundsen Bay to Ross Sea to Terra Nova Bay? On line 364 it is stated that surface dissolved metal concentrations decreased in this order, perhaps due to seasonal drawdown but I am unsure of the sampling timeline between the locations.

> [Authors' Response] Samples were not taken sequentially from the Amundsen Sea to the Ross Sea to Terra Nova Bay. Instead, the CICLOPS expedition transited from the Amundsen Sea to the Ross Sea between Station 4 and Station 22, then moved between the Ross Sea and Terra Nova Bay intermittently until the end of February. The sampling timeline between locations should be clear from the sequential numbering of stations. Thus, while some time-dependent, seasonal can be inferred, we do not claim this is the sole explanation for the regional differences observed, with the exception of the Amundsen Sea, which was only sampled early in the season.

> To clarify the expedition track, we have added a sentence to Section 2.1: "The expedition track first mapped a transect from the Amundsen Sea, through the Ross Sea, and ending in Terra Nova Bay (Stations 4–22) over 10 days from December 31, 2017 to January 9, 2018, and then sampled at stations between Terra Nova Bay and the western Ross Sea for the remainder of the expedition. "

Sections 2.2 and 2.3: Can you please clarify which samples were analyzed at sea versus the land-based laboratory? It is a little difficult to decipher at present. Importantly, were any seawater samples (and/or GSC / internal standard) analyzed both at sea and on land for intercalibration purposes?

> [Authors' Response] We have added a sentence to clarify: "Total dCo concentrations for stations 57 and 60 were analyzed in the laboratory, while all other total dCo and labile dCo concentrations were analyzed at sea."

> An internal standard was run before each sample batch, both in the laboratory and at sea. No GEOTRACES intercalibration standards, including GSC2, were run at sea.

Line 244: Shipboard incubator? Was this on deck with flow-through water and natural day/night cycle or temperature controlled with constant screened lighting? Please expand slightly.

> [Authors' Response] This sentence has been modified to read: "All incubation bottles were then sealed and placed in a flow-through shipboard incubator on the deck that exposed the incubations to a natural day/night cycle and surface-temperature seawater for 24 hours. The incubator was shielded by black mesh screening to allow 20% ambient light penetration."

Lines 359-365: This section briefly describes the pattern of decreasing surface trace metal concentrations between sampling locations and the possible mechanisms for this. Has any statistical test been performed to resolve this decreasing trend or is this assumption based on trace metal mean concentration values?

> [Authors' Response] The noted trend of decreasing surface dissolved metal concentrations are based on average concentrations and standard deviations. We believe this is clear in the text and no changes have been made.

Figure 7 clearly shows dCo:PO4 slopes are being calculated below the mixed layer. Are underlying waters consistent between the geographic sampling locations, and/or could seasonal changes in subsurface water impact slope-based stoichiometry?

> [Authors' Response] Due to the coastal location and thus highly dynamic hydrography of the samples locations, the underlying waters of each location are likely to vary by geographic region. Also, due to the vertical nature of the dissolved metal vs. $dPO_4^{3-}$ regressions in the deep ocean, only 3 of 9 slopes displayed an $R^2$ value above 0.50 (Table 3), indicating very weak linearity between dissolved metals and dissolved phosphate, and so the deep slopes are difficult to use for regional comparison.
>
> That said, the deep Amundsen Sea and deep Terra Nova Bay regions display dCo : $dPO_4^{3-}$ slopes that are more linear ($R^2 > 0.70$), and their slopes are quite similar (Amundsen Sea: $0.59 \pm 0.06$ mmol : mol, $R^2 = 0.72$; Terra Nova Bay: $0.64 \pm 0.03$ mmol : mol, $R^2 = 0.80$), indicating apparent consistency in deep cobalt stoichiometry between the underlying waters of the Amundsen Sea and Terra Nova Bay.

The metal uptake values presented in this paper are especially interesting. However, throughout the text I found inconsistent expressions of the metal uptake term ($\rho$) that made me stop and question my understanding of the text and figures. For example, the use of $\rho$Co and non-italicized ρCo for cobalt uptake rates, whilst pCo is also used to define particulate Co. The same is true for Zn and Cd. In this instance, I suggest the italicized variant for element specific uptake rates for the text, figures, and tables. Further, the use of $\rho$M, non-italicized ρM, and $\rho$TM for trace metal uptake rates. With pM also used extensively in the paper as a unit of concentration, I would suggest the italicized $\rho$M for defining trace metal uptake rates.

> [Authors' Response] Thank you for your attention on this matter – the $\rho$M (metal update) vs p (particulate) vs. pM (picomolar) terminology is unfortunately confusing. Any use of non-italicized ρCo was in error and has been corrected throughout the text, figures and tables to $\rho$Co. The one use of $\rho$TM instead of $\rho$M used in the text was an inconsistency that has been corrected.

Lines 521-524: Whilst there appears to be a clear concentration difference between CORSACS and CICLOPS mean Co profiles (Figures 8 and 9), there also appears to be considerable variability within the CICLOPS profiles. Do you have a feel for how sampling site variability within and between the cruises may have impacted the deep water dCo and labile dCo inventories shown in figure 9?

[Authors' Response] The standard deviations of the deep dCo and deep labile dCo inventories are given in Table 4 and shown as the error bars in Figure 9c. As you noted, the variability (standard deviation) of deep mean dCo concentrations were higher during the CICLOPS expedition (8 pM) compared to the CORSACS-1 (4 pM) or CORSACS-2 (6 pM) expeditions. The deep labile dCo inventories show no such differences in variability, as both show a standard deviation of 7 pM. This is an interesting point to compare, and the following sentence has been added to Sect. 4.3:

"Note that the CICLOPS expedition mean deep dCo inventory displayed a higher standard deviation (8 pM) compared to the CORSACS-1 (4 pM) and CORSACS-2 (6 pM) expeditions, indicating a higher variability of deep dCo concentration within the sites and depths sampled; no difference in standard deviation was observed within the deep labile dCo inventories of the CICLOPS and CORSACS-2 expeditions (both 7 pM)."

Lines 1015-1018: As your box model nicely shows, increased uptake, mixing and burial of DOM are all part of enhanced dCo removal from the water column, yet burial is excluded here.

[Authors' Response] We have added a sentence to the conclusion to describe the role of DOM and Co burial in the summary of the proposed mechanism. This passage now reads: "The decrease in dCo throughout the water column compared to prior observations is indicative of a multi-year mechanism, whereby the removal of dCo from the surface mixed layer via uptake over the summer has been propagated into the deep ocean via winter mixing, resulting in a decrease in dCo concentration throughout the water column. This mechanism is reliant upon increased Co uptake into organic matter and an increase in the burial rate of Co as organic matter."

**Technical corrections**

Line 42: DOM does not necessarily need to be defined here as no further use in the abstract, but rather at first mention in the main text.

[Authors' Response] The definition of the acronym POM (note: I believe this comment was intended to address POM not DOM) has been removed.

Line 128: Please provide the location of the Saito Laboratory.

[Authors' Response] This change has been made, and the passage now reads: "…supplied by the Saito laboratory (Woods Hole Oceanographic Institution; Woods Hole, MA, USA)."

Line 131: Is this a clean-air van?

[Authors' Response] Yes, the passage has been update to read: "…After deployment, the X-Niskin bottles were transported to a trace metal clean-air van and pressurized with high-purity (99.999 %) $N_2$ gas."

Line 132: Please use the term 'macronutrients' as used elsewhere in this paper.

[Authors' Response] This passage has been updated to read: "Seawater samples for macronutrients, dCo and trace metal analysis…"

Line 234: I think the reader will be interested in what resin you used for preconcentration.

[Authors' Response] Since the full ICP-MS methods for these specific samples are described elsewhere, details such as preconcentration resin type have been omitted.

Line 235: What is the elution matrix, 1 M nitric acid?

[Authors' Response] See above.

Line 242: What container/volume was the seawater used for the uptake experiments collected into from the TM rosette?

[Authors' Response] Similar to above, since the full incubation experiment methods and results for this specific data are described in detail elsewhere, methodological details have been omitted from this paper.

Line 252: Please clarify that pigment samples were collected from the trace metal CTD, as per macronutrients?

[Authors' Response] The pigment samples were not collected from the trace metal rosette, but from a standard rosette CTD deployed separately, but often within an hour or two of the trace metal rosette. This has been clarified in the text and the passage now reads: "Phytoplankton pigment samples were collected from a non-trace metal rosette deployed separately from the trace metal rosette, and were filtered and analyzed for select pigments…"

Line 299: No need to repeat (≥100 m depth) here as explained in the paragraph prior. Can just use 'deep'

[Authors' Response] The "(≥100 m depth)" in this sentence has been removed.

Line 303: Move the depths (770 m and 780 m) to line 302; i.e., elevated near-seafloor signal at Station 41 (770 m and 780 m)'…

[Authors' Response] This change has been made. The sentence now reads: "The high standard deviation of deep dCo in Terra Nova Bay is partially driven by the elevated near-seafloor signal at Station 41 (770 m and 780 m); when the two deepest points at Station 41 are omitted, the average deep dCo in Terra Nova Bay was $36 \pm 10$ pM."

Line 329: dFe has not yet been defined in the text.

[Authors' Response] dFe is now defined earlier, at the end of the introduction.

Line 359: I think the Zn concentration should be in units of nM rather than pM.

[Authors' Response] This is correct, the dZn concentration has been changed to nM, not pM.

Line 366-367: I would perhaps remove the term surface seawater, or replace with upper ocean, as the surface is generally defined as 10 m within this paper and <100 m as deep water.

[Authors' Response] The term "surface" has been removed and the passage now reads: "…uptake rates of Co, Zn and Cd within seawater collected from 0–200 m were determined…"

Line 379: Remove the word 'spikes'.

[Authors' Response] The word "spikes" has been removed.

Line 408: Depths at which an uptake rate **is** below detection…….

[Authors' Response] The word "is" has been added here.

Line 414: dCo and labile dCo **vertical** profiles…….

[Authors' Response] The word "vertical" has been added here.

Lines 507-519: Would this paragraph on the nepheloid layer be better placed within its own sub-section rather than with stoichiometry?

[Authors' Response] The reasoning for adding the nepheloid layer discussion into the stoichiometry discussion was that it decoupled dCo and $dPO_4$ ratios and thus impacted Co stoichiometry. However, I like this suggestion of a separate nepheloid layer sub-section, and I believe it will help to orient readers within this long discussion section.

The paragraph describing the nepheloid layer is now section "4.2 Elevated dCo concentrations within a benthic nepheloid layer".

Line 577: I think it may read better as; 'In contrast, the 2006 CORSACS-2 expedition reported the presence of labile dCo at five stations with concentrations of $17 \pm 7$ pM at 6 m depth and $14 \pm 9$ pM at 16 m depth, with reported labile : total dCo ratios of $0.37 \pm 0.13$ and $0.28 \pm 0.17$, respectively.'

[Authors' Response] This sentence has been updated as suggested.

Line 677: Perhaps consider moving the final paragraph on depth thresholds up to line 671. It appears to me the reader would be better served understanding how the inflection points were chosen prior to the differences between metal:PO4 slopes.

> [Authors' Response] As suggested, the sentences describing the depth threshold have been moved up, and the sentences describing the results of the depth-based analysis have been made into their own paragraph following the description of the analysis.

Line 825: Do you mean 1% lability, as per line 767?

> [Authors' Response] Yes, thank you for catching this typo. The line now reads: "…the maximum diffusion limit estimation assumed a speciation of 1% labile metals."

Line 937: 3550 pmol **dCo** m$^{-2}$ y$^{-1}$

> [Authors' Response] This change has been made and the line now reads: "A $F_{Neph}$ was calculated to be 3550 pmol dCo m$^{-2}$ d$^{-1}$ to the deep ocean…"

Line 1064: Do you mean panels d-f?

> [Authors' Response] Yes, this has been changed from "In panels d-e…" to "In panels d-f".

Table 1: The surface and deep sections of this table have different labelling for n<DL column.

> [Authors' Response] This inconsistency has been fixed.

Figure 1: The stars and circles are small and quite difficult to differentiate on paper and even more so on screen. Could these be enlarged or made clearer?

> [Authors' Response] The star symbols in Figure 1 have been enlarged.

Figure 6a: This figure would benefit from X and Y-axis lines.

> [Authors' Response] We decided not to add x and y-axis lines to this figure, as it would not follow the style of the other stoichiometry figures in this paper. However, we did correct typos in the x-axes; in subplot (a) we changed "Dissolved d$PO_4^{-3}$" to "Dissolved Phosphate" and in subplots (b-e) we changed "d$PO_4^{-3}$" to "d$PO_4^{3-}$". A similar x-axis error was updated in Figure 7.

Figure 11, Line 940: Please define the Co cycle as black arrows, as the pCo cycle has black text. Same with the pCo cycle (orange arrows).

> [Authors' Response] This correction has been made.

Table 6: Is the burial flux here calculated as dCo or pCo? Line 960 suggests pCo yet this term falls under the dCo loss parameters.

      [Authors' Response] The burial flux is calculated as particulate Co. We have changed the third table heading from "dCo loss parameters" to "Co loss parameters".